# Learning with Symmetric Label Noise: The Importance of Being Unhinged

**Brendan van Rooyen**[*,†]     **Aditya Krishna Menon**[†,*]     **Robert C. Williamson**[*,†]

[*]The Australian National University     [†]National ICT Australia
{ brendan.vanrooyen, aditya.menon, bob.williamson }@nicta.com.au

## Abstract

Convex potential minimisation is the *de facto* approach to binary classification. However, Long and Servedio [2010] proved that under symmetric label noise (SLN), minimisation of *any* convex potential over a linear function class can result in classification performance equivalent to random guessing. This ostensibly shows that convex losses are not SLN-robust. In this paper, we propose a convex, classification-calibrated loss and prove that it *is* SLN-robust. The loss avoids the Long and Servedio [2010] result by virtue of being *negatively unbounded*. The loss is a modification of the hinge loss, where one does not clamp at zero; hence, we call it the *unhinged loss*. We show that the optimal unhinged solution is equivalent to that of a strongly regularised SVM, and is the limiting solution for *any* convex potential; this implies that strong $\ell_2$ regularisation makes most standard learners SLN-robust. Experiments confirm the unhinged loss' SLN-robustness is borne out in practice. So, with apologies to Wilde [1895], while the truth is rarely pure, it *can* be simple.

## 1 Learning with symmetric label noise

Binary classification is the canonical supervised learning problem. Given an instance space $\mathcal{X}$, and samples from some distribution $D$ over $\mathcal{X} \times \{\pm 1\}$, the goal is to learn a scorer $s\colon \mathcal{X} \to \mathbb{R}$ with low *misclassification error* on future samples drawn from $D$. Our interest is in the more realistic scenario where the learner observes samples from some corruption $\bar{D}$ of $D$, where labels have some constant probability of being flipped, and the goal is still to perform well with respect to $D$. This problem is known as learning from symmetric label noise (SLN learning) [Angluin and Laird, 1988].

Long and Servedio [2010] showed that there exist linearly separable $D$ where, when the learner observes some corruption $\bar{D}$ with symmetric label noise of *any nonzero rate*, minimisation of *any convex potential* over a linear function class results in classification performance on $D$ that is equivalent to random guessing. Ostensibly, this establishes that convex losses are not "SLN-robust" and motivates the use of non-convex losses [Stempfel and Ralaivola, 2009, Masnadi-Shirazi et al., 2010, Ding and Vishwanathan, 2010, Denchev et al., 2012, Manwani and Sastry, 2013].

In this paper, we propose a convex loss and prove that it *is* SLN-robust. The loss avoids the result of Long and Servedio [2010] by virtue of being *negatively unbounded*. The loss is a modification of the hinge loss where one does not clamp at zero; thus, we call it the *unhinged loss*. This loss has several appealing properties, such as being the unique convex loss satisfying a notion of "strong" SLN-robustness (Proposition 5), being classification-calibrated (Proposition 6), consistent when minimised on $\bar{D}$ (Proposition 7), and having an simple optimal solution that is the difference of two kernel means (Equation 8). Finally, we show that this optimal solution is equivalent to that of a strongly regularised SVM (Proposition 8), and *any* twice-differentiable convex potential (Proposition 9), implying that strong $\ell_2$ regularisation endows most standard learners with SLN-robustness.

The classifier resulting from minimising the unhinged loss is not new [Devroye et al., 1996, Chapter 10], [Schölkopf and Smola, 2002, Section 1.2], [Shawe-Taylor and Cristianini, 2004, Section 5.1]. However, establishing this classifier's (strong) SLN-robustness, uniqueness thereof, and its equivalence to a highly regularised SVM solution, to our knowledge *is* novel.

## 2 Background and problem setup

Fix an instance space $\mathcal{X}$. We denote by $D$ a distribution over $\mathcal{X} \times \{\pm 1\}$, with random variables $(\mathsf{X}, \mathsf{Y}) \sim D$. Any $D$ may be expressed via the *class-conditionals* $(P, Q) = (\mathbb{P}(\mathsf{X} \mid \mathsf{Y} = 1), \mathbb{P}(\mathsf{X} \mid \mathsf{Y} = -1))$ and *base rate* $\pi = \mathbb{P}(\mathsf{Y} = 1)$, or via the *marginal* $M = \mathbb{P}(\mathsf{X})$ and *class-probability function* $\eta \colon x \mapsto \mathbb{P}(\mathsf{Y} = 1 \mid \mathsf{X} = x)$. We interchangeably write $D$ as $D_{P,Q,\pi}$ or $D_{M,\eta}$.

### 2.1 Classifiers, scorers, and risks

A *scorer* is any function $s \colon \mathcal{X} \to \mathbb{R}$. A *loss* is any function $\ell \colon \{\pm 1\} \times \mathbb{R} \to \mathbb{R}$. We use $\ell_{-1}, \ell_1$ to refer to $\ell(-1, \cdot)$ and $\ell(1, \cdot)$. The $\ell$-*conditional risk* $L_\ell \colon [0,1] \times \mathbb{R} \to \mathbb{R}$ is defined as $L_\ell \colon (\eta, v) \mapsto \eta \cdot \ell_1(v) + (1 - \eta) \cdot \ell_{-1}(v)$. Given a distribution $D$, the $\ell$-*risk* of a scorer $s$ is defined as

$$\mathbb{L}_\ell^D(s) \doteq \mathop{\mathbb{E}}_{(\mathsf{X},\mathsf{Y}) \sim D} [\ell(\mathsf{Y}, s(\mathsf{X}))], \tag{1}$$

so that $\mathbb{L}_\ell^D(s) = \mathop{\mathbb{E}}_{\mathsf{X} \sim M} [L_\ell(\eta(\mathsf{X}), s(\mathsf{X}))]$. For a set $\mathcal{S}$, $\mathbb{L}_\ell^D(\mathcal{S})$ is the set of $\ell$-risks for all scorers in $\mathcal{S}$.

A *function class* is any $\mathcal{F} \subseteq \mathbb{R}^{\mathcal{X}}$. Given some $\mathcal{F}$, the set of *restricted Bayes-optimal scorers* for a loss $\ell$ are those scorers in $\mathcal{F}$ that minimise the $\ell$-risk:

$$\mathcal{S}_\ell^{D,\mathcal{F},*} \doteq \mathop{\mathrm{Argmin}}_{s \in \mathcal{F}} \mathbb{L}_\ell^D(s).$$

The set of (unrestricted) Bayes-optimal scorers is $\mathcal{S}_\ell^{D,*} = \mathcal{S}_\ell^{D,\mathcal{F},*}$ for $\mathcal{F} = \mathbb{R}^{\mathcal{X}}$. The *restricted* $\ell$-*regret* of a scorer is its excess risk over that of any restricted Bayes-optimal scorer:

$$\mathrm{regret}_\ell^{D,\mathcal{F}}(s) \doteq \mathbb{L}_\ell^D(s) - \inf_{t \in \mathcal{F}} \mathbb{L}_\ell^D(t).$$

Binary classification is concerned with the *zero-one loss*, $\ell^{01} \colon (y, v) \mapsto [\![yv < 0]\!] + \frac{1}{2}[\![v = 0]\!]$. A loss $\ell$ is *classification-calibrated* if all its Bayes-optimal scorers are also optimal for zero-one loss: $(\forall D)\, \mathcal{S}_\ell^{D,*} \subseteq \mathcal{S}_{01}^{D,*}$. A *convex potential* is any loss $\ell \colon (y, v) \mapsto \phi(yv)$, where $\phi \colon \mathbb{R} \to \mathbb{R}_+$ is convex, non-increasing, differentiable with $\phi'(0) < 0$, and $\phi(+\infty) = 0$ [Long and Servedio, 2010, Definition 1]. All convex potentials are classification-calibrated [Bartlett et al., 2006, Theorem 2.1].

### 2.2 Learning with symmetric label noise (SLN learning)

The problem of learning with *symmetric label noise* (*SLN learning*) is the following [Angluin and Laird, 1988, Kearns, 1998, Blum and Mitchell, 1998, Natarajan et al., 2013]. For some notional "clean" distribution $D$, which we would like to observe, we instead observe samples from some corrupted distribution $\mathrm{SLN}(D, \rho)$, for some $\rho \in [0, 1/2)$. The distribution $\mathrm{SLN}(D, \rho)$ is such that the marginal distribution of instances is unchanged, but each label is independently flipped with probability $\rho$. The goal is to learn a scorer from these corrupted samples such that $\mathbb{L}_{01}^D(s)$ is small.

For any quantity in $D$, we denote its corrupted counterparts in $\mathrm{SLN}(D, \rho)$ with a bar, e.g. $\bar{M}$ for the corrupted marginal distribution, and $\bar{\eta}$ for the corrupted class-probability function; additionally, when $\rho$ is clear from context, we will occasionally refer to $\mathrm{SLN}(D, \rho)$ by $\bar{D}$. It is easy to check that the corrupted marginal distribution $\bar{M} = M$, and [Natarajan et al., 2013, Lemma 7]

$$(\forall x \in \mathcal{X})\, \bar{\eta}(x) = (1 - 2\rho) \cdot \eta(x) + \rho. \tag{2}$$

## 3 SLN-robustness: formalisation

We consider learners $(\ell, \mathcal{F})$ for a loss $\ell$ and a function class $\mathcal{F}$, with learning being the search for some $s \in \mathcal{F}$ that minimises the $\ell$-risk. Informally, $(\ell, \mathcal{F})$ is "robust" to symmetric label noise (SLN-robust) if minimising $\ell$ over $\mathcal{F}$ gives the same classifier on both the clean distribution $D$, which

the learner would *like* to observe, and $\mathrm{SLN}(D, \rho)$ for *any* $\rho \in [0, 1/2)$, which the learner *actually* observes. We now formalise this notion, and review what is known about SLN-robust learners.

## 3.1 SLN-robust learners: a formal definition

For some fixed instance space $\mathcal{X}$, let $\Delta$ denote the set of distributions on $\mathcal{X} \times \{\pm 1\}$. Given a notional "clean" distribution $D$, $\mathcal{N}_{\mathrm{sln}} \colon \Delta \to 2^{\Delta}$ returns the *set* of possible corrupted versions of $D$ the learner may observe, where labels are flipped with unknown probability $\rho$:

$$\mathcal{N}_{\mathrm{sln}} \colon D \mapsto \left\{ \mathrm{SLN}(D, \rho) \mid \rho \in \left[ 0, \frac{1}{2} \right) \right\}.$$

Equipped with this, we define our notion of SLN-robustness.

**Definition 1** (SLN-robustness). *We say that a learner $(\ell, \mathcal{F})$ is* SLN-robust *if*

$$(\forall D \in \Delta) \, (\forall \bar{D} \in \mathcal{N}_{\mathrm{sln}}(D)) \, \mathbb{L}_{01}^D(\mathcal{S}_\ell^{D,\mathcal{F},*}) = \mathbb{L}_{01}^D(\mathcal{S}_\ell^{\bar{D},\mathcal{F},*}). \tag{3}$$

That is, SLN-robustness requires that for *any* level of label noise in the observed distribution $\bar{D}$, the classification performance (wrt $D$) of the learner is the same as if the learner directly observes $D$. Unfortunately, a widely adopted class of learners is *not* SLN-robust, as we will now see.

## 3.2 Convex potentials with linear function classes are not SLN-robust

Fix $\mathcal{X} = \mathbb{R}^d$, and consider learners with a convex potential $\ell$, and a function class of linear scorers

$$\mathcal{F}_{\mathrm{lin}} = \{ x \mapsto \langle w, x \rangle \mid w \in \mathbb{R}^d \}.$$

This captures e.g. the linear SVM and logistic regression, which are widely studied in theory and applied in practice. Disappointingly, these learners are *not* SLN-robust: Long and Servedio [2010, Theorem 2] give an example where, when learning under symmetric label noise, for *any* convex potential $\ell$, the corrupted $\ell$-risk minimiser over $\mathcal{F}_{\mathrm{lin}}$ has classification performance equivalent to random guessing on $D$. This implies that $(\ell, \mathcal{F}_{\mathrm{lin}})$ is not SLN-robust[1] as per Definition 1.

**Proposition 1** (Long and Servedio [2010, Theorem 2]). *Let $\mathcal{X} = \mathbb{R}^d$ for any $d \geq 2$. Pick any convex potential $\ell$. Then, $(\ell, \mathcal{F}_{\mathrm{lin}})$ is not SLN-robust.*

## 3.3 The fallout: what learners *are* SLN-robust?

In light of Proposition 1, there are two ways to proceed in order to obtain SLN-robust learners: either we change the class of losses $\ell$, or we change the function class $\mathcal{F}$.

The first approach has been pursued in a large body of work that embraces non-convex losses [Stempfel and Ralaivola, 2009, Masnadi-Shirazi et al., 2010, Ding and Vishwanathan, 2010, Denchev et al., 2012, Manwani and Sastry, 2013]. While such losses avoid the conditions of Proposition 1, this does not automatically imply that they are SLN-robust when used with $\mathcal{F}_{\mathrm{lin}}$. In Appendix B, we present evidence that some of these losses are in fact *not* SLN-robust when used with $\mathcal{F}_{\mathrm{lin}}$.

The second approach is to consider suitably rich $\mathcal{F}$ that contains the Bayes-optimal scorer for $\bar{D}$, e.g. by employing a universal kernel. With this choice, one can still use a convex potential loss, and in fact, owing to Equation 2, *any* classification-calibrated loss.

**Proposition 2.** *Pick any classification-calibrated $\ell$. Then, $(\ell, \mathbb{R}^{\mathcal{X}})$ is SLN-robust.*

Both approaches have drawbacks. The first approach has a computational penalty, as it requires optimising a non-convex loss. The second approach has a statistical penalty, as estimation rates with a rich $\mathcal{F}$ will require a larger sample size. Thus, it appears that SLN-robustness involves a computational-statistical tradeoff. However, there is a variant of the first option: pick a loss that is convex, *but not a convex potential*. Such a loss would afford the computational and statistical advantages of minimising convex risks with linear scorers. Manwani and Sastry [2013] demonstrated that square loss, $\ell(y, v) = (1 - yv)^2$, is one such loss. We will show that there is a simpler loss that is convex and SLN-robust, but is not in the class of convex potentials by virtue of being *negatively unbounded*. To derive this loss, we first re-interpret robustness via a noise-correction procedure.

# 4 A noise-corrected loss perspective on SLN-robustness

We now re-express SLN-robustness to reason about optimal scorers on the *same distribution*, but with two *different losses*. This will help characterise a set of "strongly SLN-robust" losses.

## 4.1 Reformulating SLN-robustness via noise-corrected losses

Given any $\rho \in [0, 1/2)$, Natarajan et al. [2013, Lemma 1] showed how to associate with a loss $\ell$ a *noise-corrected* counterpart $\bar{\ell}$ such that $\mathbb{L}_\ell^D(s) = \mathbb{L}_{\bar{\ell}}^{\bar{D}}(s)$. The loss $\bar{\ell}$ is defined as follows.

**Definition 2** (Noise-corrected loss). *Given any loss $\ell$ and $\rho \in [0, 1/2)$, the noise-corrected loss $\bar{\ell}$ is*

$$(\forall y \in \{\pm 1\})\,(\forall v \in \mathbb{R})\, \bar{\ell}(y, v) = \frac{(1 - \rho) \cdot \ell(y, v) - \rho \cdot \ell(-y, v)}{1 - 2\rho}. \tag{4}$$

Since $\bar{\ell}$ depends on the unknown parameter $\rho$, it is not directly usable to design an SLN-robust learner. Nonetheless, it is a useful theoretical device, since, by construction, for *any* $\mathcal{F}$, $\mathcal{S}_\ell^{D,\mathcal{F},*} = \mathcal{S}_{\bar{\ell}}^{\bar{D},\mathcal{F},*}$. This means that a sufficient condition for $(\ell, \mathcal{F})$ to be SLN-robust is for $\mathcal{S}_\ell^{\bar{D},\mathcal{F},*} = \mathcal{S}_{\bar{\ell}}^{\bar{D},\mathcal{F},*}$. Ghosh et al. [2015, Theorem 1] proved a *sufficient* condition on $\ell$ such that this holds, namely,

$$(\exists C \in \mathbb{R})(\forall v \in \mathbb{R})\, \ell_1(v) + \ell_{-1}(v) = C. \tag{5}$$

Interestingly, Equation 5 is *necessary* for a *stronger* notion of robustness, which we now explore.

## 4.2 Characterising a stronger notion of SLN-robustness

As the first step towards a stronger notion of robustness, we rewrite (with a slight abuse of notation)

$$\mathbb{L}_\ell^D(s) = \underset{(\mathsf{X},\mathsf{Y}) \sim D}{\mathbb{E}}[\ell(\mathsf{Y}, s(\mathsf{X}))] = \underset{(\mathsf{Y},\mathsf{S}) \sim R(D,s)}{\mathbb{E}}[\ell(\mathsf{Y}, \mathsf{S})] \doteq \mathbb{L}_\ell(R(D, s)),$$

where $R(D, s)$ is a distribution over labels and *scores*. Standard SLN-robustness requires that label noise does not change the $\ell$-risk minimisers, i.e. that if $s$ is such that $\mathbb{L}_\ell(R(D, s)) \leq \mathbb{L}_\ell(R(D, s'))$ for all $s'$, the same relation holds with $\bar{D}$ in place of $D$. Strong SLN-robustness strengthens this notion by requiring that label noise does not affect the ordering of *all* pairs of joint distributions over labels and scores. (This of course trivially implies SLN-robustness.) As with the definition of $\bar{D}$, given a distribution $R$ over labels and scores, let $\bar{R}$ be the corresponding distribution where labels are flipped with probability $\rho$. Strong SLN-robustness can then be made precise as follows.

**Definition 3** (Strong SLN-robustness). *Call a loss $\ell$ strongly SLN-robust if for every $\rho \in [0, 1/2)$,*

$$(\forall R, R')\, \mathbb{L}_\ell(R) \leq \mathbb{L}_\ell(R') \iff \mathbb{L}_\ell(\bar{R}) \leq \mathbb{L}_\ell(\bar{R}').$$

We now re-express strong SLN-robustness using a notion of *order equivalence* of loss pairs, which simply requires that two losses order all distributions over labels and scores identically.

**Definition 4** (Order equivalent loss pairs). *Call a pair of losses $(\ell, \tilde{\ell})$ order equivalent if*

$$(\forall R, R')\, \mathbb{L}_\ell(R) \leq \mathbb{L}_\ell(R') \iff \mathbb{L}_{\tilde{\ell}}(R) \leq \mathbb{L}_{\tilde{\ell}}(R').$$

Clearly, order equivalence of $(\ell, \bar{\ell})$ implies $\mathcal{S}_\ell^{D,\mathcal{F},*} = \mathcal{S}_{\bar{\ell}}^{D,\mathcal{F},*}$, which in turn implies SLN-robustness. It is thus not surprising that we can relate order equivalence to strong SLN-robustness of $\ell$.

**Proposition 3.** *A loss $\ell$ is strongly SLN-robust iff for every $\rho \in [0, 1/2)$, $(\ell, \bar{\ell})$ are order equivalent.*

This connection now lets us exploit a classical result in decision theory about order equivalent losses being affine transformations of each other. Combined with the definition of $\bar{\ell}$, this lets us conclude that the sufficient condition of Equation 5 is also *necessary* for strong SLN-robustness of $\ell$.

**Proposition 4.** *A loss $\ell$ is strongly SLN-robust if and only if it satisfies Equation 5.*

We now return to our original goal, which was to find a convex $\ell$ that is SLN-robust for $\mathcal{F}_{\text{lin}}$ (and ideally more general function classes). The above suggests that to do so, it is reasonable to consider those losses that satisfy Equation 5. Unfortunately, it is evident that if $\ell$ is convex, non-constant, and bounded below by zero, then it cannot possibly be admissible in this sense. But we now show that removing the boundedness restriction allows for the existence of a convex admissible loss.

# 5 The unhinged loss: a convex, strongly SLN-robust loss

Consider the following simple, but non-standard convex loss:

$$\ell_1^{\mathrm{unh}}(v) = 1 - v \text{ and } \ell_{-1}^{\mathrm{unh}}(v) = 1 + v.$$

Compared to the hinge loss, the loss does not clamp at zero, i.e. it does not have a hinge. (Thus, peculiarly, it is negatively unbounded, an issue we discuss in §5.3.) Thus, we call this the *unhinged loss*[2]. The loss has a number of attractive properties, the most immediate being is its SLN-robustness.

## 5.1 The unhinged loss is strongly SLN-robust

Since $\ell_1^{\mathrm{unh}}(v) + \ell_{-1}^{\mathrm{unh}}(v) = 2$, Proposition 4 implies that $\ell^{\mathrm{unh}}$ is strongly SLN-robust, and thus that $(\ell^{\mathrm{unh}}, \mathcal{F})$ is SLN-robust for *any* $\mathcal{F}$. Further, the following uniqueness property is not hard to show.

**Proposition 5.** *Pick any convex loss $\ell$. Then,*

$$(\exists C \in \mathbb{R}) \, \ell_1(v) + \ell_{-1}(v) = C \iff (\exists A, B, D \in \mathbb{R}) \, \ell_1(v) = -A \cdot v + B, \ell_{-1}(v) = A \cdot v + D.$$

*That is, up to scaling and translation, $\ell^{\mathrm{unh}}$ is the only convex loss that is strongly SLN-robust.*

Returning to the case of linear scorers, the above implies that $(\ell^{\mathrm{unh}}, \mathcal{F}_{\mathrm{lin}})$ is SLN-robust. This does not contradict Proposition 1, since $\ell^{\mathrm{unh}}$ is not a convex potential as it is negatively unbounded. Intuitively, this property allows the loss to offset the penalty incurred by instances that are misclassified with high margin by awarding a "gain" for instances that correctly classified with high margin.

## 5.2 The unhinged loss is classification calibrated

SLN-robustness is by itself insufficient for a learner to be useful. For example, a loss that is uniformly zero is strongly SLN-robust, but is useless as it is not classification-calibrated. Fortunately, the unhinged loss is classification-calibrated, as we now establish. For technical reasons (see §5.3), we operate with $\mathcal{F}_B = [-B, +B]^{\mathcal{X}}$, the set of scorers with range bounded by $B \in [0, \infty)$.

**Proposition 6.** *Fix $\ell = \ell^{\mathrm{unh}}$. For any $D_{M,\eta}$, $B \in [0, \infty)$, $\mathcal{S}_\ell^{D, \mathcal{F}_B, *} = \{x \mapsto B \cdot \mathrm{sign}(2\eta(x) - 1)\}$.*

Thus, for every $B \in [0, \infty)$, the restricted Bayes-optimal scorer over $\mathcal{F}_B$ has the same sign as the Bayes-optimal classifier for 0-1 loss. In the limiting case where $\mathcal{F} = \mathbb{R}^{\mathcal{X}}$, the optimal scorer is attainable if we operate over the extended reals $\mathbb{R} \cup \{\pm\infty\}$, so that $\ell^{\mathrm{unh}}$ is classification-calibrated.

## 5.3 Enforcing boundedness of the loss

While the classification-calibration of $\ell^{\mathrm{unh}}$ is encouraging, Proposition 6 implies that its (unrestricted) Bayes-risk is $-\infty$. Thus, the regret of every non-optimal scorer $s$ is identically $+\infty$, which hampers analysis of consistency. In orthodox decision theory, analogous theoretical issues arise when attempting to establish basic theorems with unbounded losses [Ferguson, 1967, pg. 78].

We can side-step this issue by restricting attention to bounded scorers, so that $\ell^{\mathrm{unh}}$ is effectively bounded. By Proposition 6, this does not affect the classification-calibration of the loss. In the context of linear scorers, boundedness of scorers can be achieved by regularisation: instead of working with $\mathcal{F}_{\mathrm{lin}}$, one can instead use $\mathcal{F}_{\mathrm{lin},\lambda} = \{x \mapsto \langle w, x \rangle \mid ||w||_2 \leq 1/\sqrt{\lambda}\}$, where $\lambda > 0$, so that $\mathcal{F}_{\mathrm{lin},\lambda} \subseteq \mathcal{F}_{R/\sqrt{\lambda}}$ for $R = \sup_{x \in \mathcal{X}} ||x||_2$. Observe that as $(\ell^{\mathrm{unh}}, \mathcal{F})$ is SLN-robust for *any* $\mathcal{F}$, $(\ell^{\mathrm{unh}}, \mathcal{F}_{\mathrm{lin},\lambda})$ is SLN-robust for any $\lambda > 0$. As we shall see in §6.3, working with $\mathcal{F}_{\mathrm{lin},\lambda}$ also lets us establish SLN-robustness of the hinge loss when $\lambda$ is large.

## 5.4 Unhinged loss minimisation on corrupted distribution is consistent

Using bounded scorers makes it possible to establish a surrogate regret bound for the unhinged loss. This shows classification consistency of unhinged loss minimisation on the *corrupted* distribution.

**Proposition 7.** *Fix $\ell = \ell^{\mathrm{unh}}$. Then, for any $D, \rho \in [0, 1/2)$, $B \in [1, \infty)$, and scorer $s \in \mathcal{F}_B$,*

$$\mathrm{regret}_{01}^D(s) \leq \mathrm{regret}_{\ell}^{D, \mathcal{F}_B}(s) = \frac{1}{1 - 2\rho} \cdot \mathrm{regret}_{\ell}^{\bar{D}, \mathcal{F}_B}(s).$$

Standard rates of convergence via generalisation bounds are also trivial to derive; see the Appendix.

# 6 Learning with the unhinged loss and kernels

We now show that the optimal solution for the unhinged loss when employing regularisation and kernelised scorers has a simple form. This sheds further light on SLN-robustness and regularisation.

## 6.1 The centroid classifier optimises the unhinged loss

Consider minimising the unhinged risk over the class of kernelised scorers $\mathcal{F}_{\mathcal{H}, \lambda} = \{s \colon x \mapsto \langle w, \Phi(x) \rangle_{\mathcal{H}} \mid \|w\|_{\mathcal{H}} \leq 1/\sqrt{\lambda}\}$ for some $\lambda > 0$, where $\Phi \colon \mathcal{X} \to \mathcal{H}$ is a feature mapping into a reproducing kernel Hilbert space $\mathcal{H}$ with kernel $k$. Equivalently, given a distribution[3] $D$, we want

$$w_{\mathrm{unh}, \lambda}^* = \underset{w \in \mathcal{H}}{\mathrm{argmin}} \ \underset{(\mathsf{X}, \mathsf{Y}) \sim D}{\mathbb{E}} [1 - \mathsf{Y} \cdot \langle w, \Phi(\mathsf{X}) \rangle] + \frac{\lambda}{2} \langle w, w \rangle_{\mathcal{H}}. \tag{6}$$

The first-order optimality condition implies that

$$w_{\mathrm{unh}, \lambda}^* = \frac{1}{\lambda} \cdot \underset{(\mathsf{X}, \mathsf{Y}) \sim D}{\mathbb{E}} [\mathsf{Y} \cdot \Phi(\mathsf{X})], \tag{7}$$

which is the *kernel mean map* of $D$ [Smola et al., 2007], and thus the optimal unhinged scorer is

$$s_{\mathrm{unh}, \lambda}^* \colon x \mapsto \frac{1}{\lambda} \cdot \underset{(\mathsf{X}, \mathsf{Y}) \sim D}{\mathbb{E}} [\mathsf{Y} \cdot k(\mathsf{X}, x)] = x \mapsto \frac{1}{\lambda} \cdot \left( \pi \cdot \underset{\mathsf{X} \sim P}{\mathbb{E}} [k(\mathsf{X}, x)] - (1 - \pi) \cdot \underset{\mathsf{X} \sim Q}{\mathbb{E}} [k(\mathsf{X}, x)] \right). \tag{8}$$

From Equation 8, the unhinged solution is equivalent to a *nearest centroid classifier* [Manning et al., 2008, pg. 181] [Tibshirani et al., 2002] [Shawe-Taylor and Cristianini, 2004, Section 5.1]. Equation 8 gives a simple way to understand the SLN-robustness of $(\ell^{\mathrm{unh}}, \mathcal{F}_{\mathcal{H}, \lambda})$, as the optimal scorers on the clean and corrupted distributions only differ by a scaling (see the Appendix):

$$(\forall x \in \mathcal{X}) \ \underset{(\mathsf{X}, \mathsf{Y}) \sim D}{\mathbb{E}} [\mathsf{Y} \cdot k(\mathsf{X}, x)] = \frac{1}{1 - 2\rho} \cdot \underset{(\mathsf{X}, \bar{\mathsf{Y}}) \sim \bar{D}}{\mathbb{E}} [\bar{\mathsf{Y}} \cdot k(\mathsf{X}, x)]. \tag{9}$$

Interestingly, Servedio [1999, Theorem 4] established that a nearest centroid classifier (which they termed "AVERAGE") is robust to a general class of label noise, but required the assumption that $M$ is uniform over the unit sphere. Our result establishes that SLN robustness of the classifier holds without any assumptions on $M$. In fact, Ghosh et al. [2015, Theorem 1] lets one quantify the unhinged loss' performance under a more general noise model; see the Appendix for discussion.

## 6.2 Practical considerations

We note several points relating to practical usage of the unhinged loss with kernelised scorers. First, cross-validation is not required to select $\lambda$, since changing $\lambda$ only changes the magnitude of scores, *not their sign*. Thus, for the purposes of classification, one can simply use $\lambda = 1$.

Second, we can easily extend the scorers to use a bias regularised with strength $0 < \lambda_b \neq \lambda$. Tuning $\lambda_b$ is equivalent to computing $s_{\mathrm{unh}, \lambda}^*$ as per Equation 8, and tuning a threshold on a holdout set.

Third, when $\mathcal{H} = \mathbb{R}^d$ for $d$ small, we can store $w_{\mathrm{unh}, \lambda}^*$ explicitly, and use this to make predictions. For high (or infinite) dimensional $\mathcal{H}$, we can either make predictions directly via Equation 8, or use random Fourier features [Rahimi and Recht, 2007] to (approximately) embed $\mathcal{H}$ into some low-dimensional $\mathbb{R}^d$, and then store $w_{\mathrm{unh}, \lambda}^*$ as usual. (The latter requires a translation-invariant kernel.)

We now show that under some assumptions, $w_{\mathrm{unh}, \lambda}^*$ coincides with the solution of two established methods; the Appendix discusses some further relationships, e.g. to the maximum mean discrepancy.

## 6.3 Equivalence to a highly regularised SVM and other convex potentials

There is an interesting equivalence between the unhinged solution and that of a *highly regularised SVM*. This has been noted in e.g. Hastie et al. [2004, Section 6], which showed how SVMs approach a nearest centroid classifier, which is of course the optimal unhinged solution.

**Proposition 8.** *Pick any $D$ and $\Phi\colon \mathcal{X} \to \mathcal{H}$ with $R = \sup_{x \in \mathcal{X}} ||\Phi(x)||_{\mathcal{H}} < \infty$. For any $\lambda > 0$, let*

$$w_{\text{hinge},\lambda}^* = \operatorname*{argmin}_{w \in \mathcal{H}} \mathop{\mathbb{E}}_{(\mathsf{X},\mathsf{Y}) \sim D} [\max(0, 1 - \mathsf{Y} \cdot \langle w, \Phi(x) \rangle_{\mathcal{H}})] + \frac{\lambda}{2} \langle w, w \rangle_{\mathcal{H}}$$

*be the soft-margin SVM solution. Then, if $\lambda \geq R^2$, $w_{\text{hinge},\lambda}^* = w_{\text{unh},\lambda}^*$.*

Since $(\ell^{\text{unh}}, \mathcal{F}_{\mathcal{H},\lambda})$ is SLN-robust, it follows that for $\ell^{\text{hinge}}\colon (y,v) \mapsto \max(0, 1-yv)$, $(\ell^{\text{hinge}}, \mathcal{F}_{\mathcal{H},\lambda})$ is similarly SLN-robust *provided $\lambda$ is sufficiently large*. That is, strong $\ell_2$ regularisation (and a bounded feature map) endows the hinge loss with SLN-robustness[4]. Proposition 8 can be generalised to show that $w_{\text{unh},\lambda}^*$ is the limiting solution of *any* twice differentiable convex potential. This shows that *strong $\ell_2$ regularisation endows most learners with SLN-robustness*. Intuitively, with strong regularisation, one only considers the behaviour of a loss near zero; since a convex potential $\phi$ has $\phi'(0) < 0$, it will behave similarly to its linear approximation around zero, viz. the unhinged loss.

**Proposition 9.** *Pick any $D$, bounded feature mapping $\Phi\colon \mathcal{X} \to \mathcal{H}$, and twice differentiable convex potential $\phi$ with $\phi''([-1,1])$ bounded. Let $w_{\phi,\lambda}^*$ be the minimiser of the regularised $\phi$ risk. Then,*

$$\lim_{\lambda \to \infty} \left\| \frac{w_{\phi,\lambda}^*}{||w_{\phi,\lambda}^*||_{\mathcal{H}}} - \frac{w_{\text{unh},\lambda}^*}{||w_{\text{unh},\lambda}^*||_{\mathcal{H}}} \right\|_{\mathcal{H}}^2 = 0.$$

## 6.4 Equivalence to Fisher Linear Discriminant with whitened data

For binary classification on $D_{M,\eta}$, the Fisher Linear Discriminant (FLD) finds a weight vector proportional to the minimiser of square loss $\ell^{\text{sq}}\colon (y,v) \mapsto (1-yv)^2$ [Bishop, 2006, Section 4.1.5],

$$w_{\text{sq},\lambda}^* = (\mathbb{E}_{\mathsf{X} \sim M}[\mathsf{X}\mathsf{X}^T] + \lambda I)^{-1} \cdot \mathbb{E}_{(\mathsf{X},\mathsf{Y}) \sim D}[\mathsf{Y} \cdot \mathsf{X}]. \qquad (10)$$

By Equation 9, and the fact that the corrupted marginal $\bar{M} = M$, $w_{\text{sq},\lambda}^*$ is only changed by a scaling factor under label noise. This provides an alternate proof of the fact that $(\ell^{\text{sq}}, \mathcal{F}_{\text{lin}})$ is SLN-robust[5] [Manwani and Sastry, 2013, Theorem 2]. Clearly, the unhinged loss solution $w_{\text{unh},\lambda}^*$ is equivalent to the FLD and square loss solution $w_{\text{sq},\lambda}^*$ when the input data is whitened i.e. $\mathbb{E}_{\mathsf{X} \sim M} [\mathsf{X}\mathsf{X}^T] = I$. With a well-specified $\mathcal{F}$, e.g. with a universal kernel, both the unhinged and square loss asymptotically recover the optimal classifier, but the unhinged loss does not require a matrix inversion. With a misspecified $\mathcal{F}$, one cannot in general argue for the superiority of the unhinged loss over square loss, or vice-versa, as there is no universally good surrogate to the 0-1 loss [Reid and Williamson, 2010, Appendix A]; the Appendix illustrate examples where both losses may underperform.

## 7 SLN-robustness of unhinged loss: empirical illustration

We now illustrate that the unhinged loss' SLN-robustness is empirically manifest. We reiterate that with high regularisation, the unhinged solution is equivalent to an SVM (and in the limit any classification-calibrated loss) solution. Thus, we do *not* aim to assert that the unhinged loss is "better" than other losses, but rather, to demonstrate that its SLN-robustness is not *purely* theoretical.

We first show that the unhinged risk minimiser performs well on the example of Long and Servedio [2010] (henceforth LS10). Figure 1 shows the distribution $D$, where $\mathcal{X} = \{(1,0), (\gamma, 5\gamma), (\gamma, -\gamma)\} \subset \mathbb{R}^2$, with marginal distribution $M = \{\frac{1}{4}, \frac{1}{4}, \frac{1}{2}\}$ and all three instances are deterministically positive. We pick $\gamma = 1/2$. The unhinged minimiser perfectly classifies all three points, regardless of the level of label noise (Figure 1). The hinge minimiser is perfect when there is no noise, but with even a small amount of noise, achieves a 50% error rate.

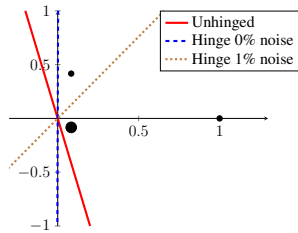

Figure 1: `LS10` dataset.

| | Hinge | $t$-logistic | Unhinged |
|---|---|---|---|
| $\rho = 0$ | $0.00 \pm 0.00$ | $0.00 \pm 0.00$ | $0.00 \pm 0.00$ |
| $\rho = 0.1$ | $0.15 \pm 0.27$ | $0.00 \pm 0.00$ | $0.00 \pm 0.00$ |
| $\rho = 0.2$ | $0.21 \pm 0.30$ | $0.00 \pm 0.00$ | $0.00 \pm 0.00$ |
| $\rho = 0.3$ | $0.38 \pm 0.37$ | $0.22 \pm 0.08$ | $0.00 \pm 0.00$ |
| $\rho = 0.4$ | $0.42 \pm 0.36$ | $0.22 \pm 0.08$ | $0.00 \pm 0.00$ |
| $\rho = 0.49$ | $0.47 \pm 0.38$ | $0.39 \pm 0.23$ | $0.34 \pm 0.48$ |

Table 1: Mean and standard deviation of the 0-1 error over 125 trials on `LS10`. Grayed cells denote the best performer at that noise rate.

We next consider empirical risk minimisers from a random training sample: we construct a training set of 800 instances, injected with varying levels of label noise, and evaluate classification performance on a test set of 1000 instances. We compare the hinge, $t$-logistic (for $t = 2$) [Ding and Vishwanathan, 2010] and unhinged minimisers using a linear scorer *without* a bias term, and regularisation strength $\lambda = 10^{-16}$. From Table 1, even at 40% label noise, the unhinged classifier is able to find a perfect solution. By contrast, both other losses suffer at even moderate noise rates.

We next report results on some UCI datasets, where we additionally tune a threshold so as to ensure the best training set 0-1 accuracy. Table 2 summarises results on a sample of four datasets. (The Appendix contains results with more datasets, performance metrics, and losses.) Even at noise close to 50%, the unhinged loss is often able to learn a classifier with some discriminative power.

| | Hinge | $t$-Logistic | Unhinged |
|---|---|---|---|
| $\rho = 0$ | $0.00 \pm 0.00$ | $0.00 \pm 0.00$ | $0.00 \pm 0.00$ |
| $\rho = 0.1$ | $0.01 \pm 0.03$ | $0.01 \pm 0.03$ | $0.00 \pm 0.00$ |
| $\rho = 0.2$ | $0.06 \pm 0.12$ | $0.04 \pm 0.05$ | $0.00 \pm 0.01$ |
| $\rho = 0.3$ | $0.17 \pm 0.20$ | $0.09 \pm 0.11$ | $0.02 \pm 0.07$ |
| $\rho = 0.4$ | $0.35 \pm 0.24$ | $0.24 \pm 0.16$ | $0.13 \pm 0.22$ |
| $\rho = 0.49$ | $0.60 \pm 0.20$ | $0.49 \pm 0.20$ | $0.45 \pm 0.33$ |

(a) `iris`.

| | Hinge | $t$-Logistic | Unhinged |
|---|---|---|---|
| $\rho = 0$ | $0.05 \pm 0.00$ | $0.05 \pm 0.00$ | $0.05 \pm 0.00$ |
| $\rho = 0.1$ | $0.06 \pm 0.01$ | $0.07 \pm 0.02$ | $0.05 \pm 0.00$ |
| $\rho = 0.2$ | $0.06 \pm 0.01$ | $0.08 \pm 0.03$ | $0.05 \pm 0.00$ |
| $\rho = 0.3$ | $0.08 \pm 0.04$ | $0.11 \pm 0.05$ | $0.05 \pm 0.01$ |
| $\rho = 0.4$ | $0.14 \pm 0.10$ | $0.24 \pm 0.13$ | $0.09 \pm 0.10$ |
| $\rho = 0.49$ | $0.45 \pm 0.26$ | $0.49 \pm 0.16$ | $0.46 \pm 0.30$ |

(b) `housing`.

| | Hinge | $t$-Logistic | Unhinged |
|---|---|---|---|
| $\rho = 0$ | $0.00 \pm 0.00$ | $0.00 \pm 0.00$ | $0.00 \pm 0.00$ |
| $\rho = 0.1$ | $0.10 \pm 0.08$ | $0.11 \pm 0.02$ | $0.00 \pm 0.00$ |
| $\rho = 0.2$ | $0.19 \pm 0.11$ | $0.15 \pm 0.02$ | $0.00 \pm 0.00$ |
| $\rho = 0.3$ | $0.31 \pm 0.13$ | $0.22 \pm 0.03$ | $0.01 \pm 0.00$ |
| $\rho = 0.4$ | $0.39 \pm 0.13$ | $0.33 \pm 0.04$ | $0.02 \pm 0.02$ |
| $\rho = 0.49$ | $0.50 \pm 0.16$ | $0.48 \pm 0.04$ | $0.34 \pm 0.21$ |

(c) `usps0v7`.

| | Hinge | $t$-Logistic | Unhinged |
|---|---|---|---|
| $\rho = 0$ | $0.05 \pm 0.00$ | $0.04 \pm 0.00$ | $0.19 \pm 0.00$ |
| $\rho = 0.1$ | $0.15 \pm 0.03$ | $0.24 \pm 0.00$ | $0.19 \pm 0.01$ |
| $\rho = 0.2$ | $0.21 \pm 0.03$ | $0.24 \pm 0.00$ | $0.19 \pm 0.01$ |
| $\rho = 0.3$ | $0.25 \pm 0.03$ | $0.24 \pm 0.00$ | $0.19 \pm 0.03$ |
| $\rho = 0.4$ | $0.31 \pm 0.05$ | $0.24 \pm 0.00$ | $0.22 \pm 0.05$ |
| $\rho = 0.49$ | $0.48 \pm 0.09$ | $0.40 \pm 0.24$ | $0.45 \pm 0.08$ |

(d) `splice`.

Table 2: Mean and standard deviation of the 0-1 error over 125 trials on UCI datasets.

## 8 Conclusion and future work

We proposed a convex, classification-calibrated loss, proved that is robust to symmetric label noise (SLN-robust), showed it is the unique loss that satisfies a notion of strong SLN-robustness, established that it is optimised by the nearest centroid classifier, and showed that most convex potentials, such as the SVM, are also SLN-robust when highly regularised. So, with apologies to Wilde [1895]:

> While the truth is rarely pure, it *can* be simple.

**Acknowledgments**

NICTA is funded by the Australian Government through the Department of Communications and the Australian Research Council through the ICT Centre of Excellence Program. The authors thank Cheng Soon Ong for valuable comments on a draft of this paper.

## Footnotes

[1] Even if we were content with a difference of $\epsilon \in [0, 1/2]$ between the clean and corrupted minimisers' performance, Long and Servedio [2010, Theorem 2] implies that in the worst case $\epsilon = 1/2$.

[2]This loss has been considered in Sriperumbudur et al. [2009], Reid and Williamson [2011] in the context of maximum mean discrepancy; see the Appendix. The analysis of its SLN-robustness is to our knowledge novel.

[3]Given a training sample $\mathsf{S} \sim D^n$, we can use plugin estimates as appropriate.

[4] Long and Servedio [2010, Section 6] show that $\ell_1$ regularisation does not endow SLN-robustness.

[5] Square loss escapes the result of Long and Servedio [2010] since it is not monotone decreasing.

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
