[Supplementary Material]

# Proofs for "Learning with Symmetric Label Noise: The Importance of Being Unhinged"

## A   Proofs of results in main body

We now present proofs of all results in the main body.

*Proof of Proposition 1.* This result is implicit in Long and Servedio [2010, Theorem 2]; the aim of this proof is simply to make the result explicit, and to cast it in our terminology.

Let $\mathfrak{X} = \{(1,0), (\gamma, 5\gamma), (\gamma, -\gamma), (\gamma, -\gamma)\} \subset \mathbb{R}^2$, for some $\gamma < 1/6$. Define a distribution $D$ as follows: let the marginal distribution over $\mathfrak{X}$ be uniform, and let $\eta \colon x \mapsto 1$, i.e. every example is deterministically positive.

Now suppose we observe some $\mathrm{SLN}(D, \rho)$, for $\rho \in [0, 1/2)$. We minimise the $\ell$-risk some convex potential $\ell \colon (y, v) \mapsto \phi(y, v)$ using a linear function class[6] $\mathcal{F}_{\mathrm{lin}}$. Then, Long and Servedio [2010, Theorem 2] establishes that

$$(\forall s \in \mathcal{S}_\ell^{\bar{D}, \mathcal{F}_{\mathrm{lin}}, *}) \, \mathbb{L}_{01}^D(s) = \frac{1}{2}.$$

On the other hand, since $D$ is linearly separable and a convex potential $\ell$ is classification-calibrated, we must have $\mathbb{L}_{01}^D(\mathcal{S}_\ell^{D, \mathcal{F}_{\mathrm{lin}}, *}) = 0$. Consequently, for any convex potential $\ell$, $(\ell, \mathcal{F}_{\mathrm{lin}})$ is not SLN-robust. □

*Proof of Proposition 2.* Let $\bar{\eta}$ be the class-probability function of $\bar{D}$. By [Natarajan et al., 2013, Lemma 7],

$$(\forall x \in \mathfrak{X}) \, \mathrm{sign}(2\bar{\eta}(x) - 1) = \mathrm{sign}(2\eta(x) - 1),$$

so that the optimal classifiers on the clean and corrupted distributions coincide. Therefore, intuitively, if the Bayes-optimal solution for loss recovers $\mathrm{sign}(2\bar{\eta}(x) - 1)$, it will also recover $\mathrm{sign}(2\eta(x) - 1)$. Formally, since $\ell$ is classification-calibrated, for any $D \in \Delta$, and $s \in \mathcal{S}_\ell^{D, *}$

$$(\forall x \in \mathfrak{X}) \, \mathrm{sign}(s(x)) = \mathrm{sign}(2\eta(x) - 1),$$

and similarly, for any $\bar{D} \in \mathcal{N}_{\mathrm{sln}}(D)$, and $\bar{s} \in \mathcal{S}_\ell^{\bar{D}, *}$

$$(\forall x \in \mathfrak{X}) \, \mathrm{sign}(\bar{s}(x)) = \mathrm{sign}(2\bar{\eta}(x) - 1).$$

Thus, for any $D, \bar{D}$, since the 0-1 risk of a scorer depends only on its sign,

$$\begin{aligned}
\mathbb{L}_{01}^D(s) &= \mathbb{L}_{01}^D(\mathrm{sign}(s)) \\
&= \mathbb{L}_{01}^D(\mathrm{sign}(2\eta - 1)) \\
&= \mathbb{L}_{01}^D(\mathrm{sign}(2\bar{\eta} - 1)) \\
&= \mathbb{L}_{01}^D(\mathrm{sign}(\bar{s})) \\
&= \mathbb{L}_{01}^D(\bar{s}).
\end{aligned}$$

Consequently, $(\ell, \mathbb{R}^{\mathfrak{X}})$ is SLN-robust. □

*Proof of Proposition 3.* For brevity, we will write $R \preceq_\ell R'$ to mean that $\mathbb{L}_\ell(R) \leq \mathbb{L}_\ell(R')$, so that strong SLN-robustness is

$$(\forall \rho \in [0, 1/2)) \, (\forall R, R') \, R \preceq_\ell R' \iff \bar{R} \preceq_\ell \bar{R}'.$$

while order equivalence is

$$(\forall R, R') \, R \preceq_\ell R' \iff R \preceq_{\tilde{\ell}} R'.$$

Observe also that by definition of $\bar{\ell}$ and $\bar{R}$,

$$\mathbb{L}_\ell(R) = \mathop{\mathbb{E}}_{(Y,S)\sim R}\left[\ell(Y,S)\right] = \mathop{\mathbb{E}}_{(Y,\bar{S})\sim\bar{R}}\left[\bar{\ell}(Y,\bar{S})\right] = \mathbb{L}_{\bar{\ell}}(\bar{R}).$$

( $\impliedby$ ). Pick any $\rho \in [0, 1/2)$. Suppose that $(\ell, \bar{\ell})$ are order equivalent. We have

$$(\forall R, R')\, R \preceq_\ell R' \iff \bar{R} \preceq_{\bar{\ell}} \bar{R}'$$
$$\iff \bar{R} \preceq_\ell \bar{R}',$$

where the first line is from definition of $\bar{\ell}$, and the second is by assumed order equivalence of $(\ell, \bar{\ell})$. As $\rho$ is arbitrary, we conclude that $\ell$ is strongly SLN-robust.

( $\implies$ ). Pick any $\rho \in [0, 1/2)$. For this direction, we will need to define the following "inverse" noise-corrected loss,

$$(\forall y \in \{\pm 1\})\,(\forall v \in \mathbb{R})\, \tilde{\ell}(y, v) = (1 - \rho) \cdot \ell(y, v) + \rho \cdot \ell(-y, v)$$

which is so named because $\tilde{\bar{\ell}} = \ell$.

Now suppose that $\ell$ is strongly SLN-robust. We have by definition

$$(\forall R, R')\, R \preceq_\ell R' \iff \bar{R} \preceq_\ell \bar{R}'$$
$$\iff R \preceq_{\tilde{\ell}} R',$$

where the second line by the fact that $\mathbb{L}_{\bar{\ell}}^{\bar{D}}(s) = \mathbb{L}_{\tilde{\ell}}^D(s)$. This means that $(\ell, \tilde{\ell})$ are order equivalent for any $\rho \in [0, 1/2)$. Thus, by Lemma 10,

$$(\forall \rho \in [0, 1/2))\,(\forall y \in \{\pm 1\})\,(\forall v \in \mathbb{R})\, \tilde{\ell}(y, v) = \alpha \cdot \ell(y, v) + \beta.$$

But now using the fact that $\tilde{\bar{\ell}} = \ell$, we get

$$(\forall \rho \in [0, 1/2))\,(\forall y \in \{\pm 1\})\,(\forall v \in \mathbb{R})\, \ell(y, v) = \alpha \cdot \bar{\ell}(y, v) + \beta.$$

Applying Lemma 10 once more, we conclude that $(\ell, \bar{\ell})$ are order equivalent for any $\rho \in [0, 1/2)$. $\quad\square$

*Proof of Proposition 4.* ( $\impliedby$ ). For standard SLN-robustness, this is shown in Ghosh et al. [2015, Theorem 1]; for strong SLN-robustness, the same basic proof strategy is adopted. Suppose that Equation 5 holds. Then,

$$\begin{aligned}
\bar{\ell}(y, v) &= \frac{(1 - \rho) \cdot \ell(y, v) - \rho \cdot \ell(-y, v)}{1 - 2\rho} \\
&= \frac{(1 - \rho) \cdot \ell(y, v) - \rho \cdot (C - \ell(y, v))}{1 - 2\rho} \\
&= \frac{\ell(y, v) - \rho \cdot C}{1 - 2\rho},
\end{aligned}$$

at which stage we appeal to Lemma 10 to conclude that $(\ell, \bar{\ell})$ are order equivalent for any $\rho$. Thus by Proposition 3, $\ell$ is strongly SLN-robust.

( $\implies$ ). We have shown in Proposition 3 that strong SLN-robustness is equivalent to order equivalence of $(\ell, \bar{\ell})$ for every $\rho \in [0, 1/2)$. Thus, by Lemma 10,

$$(\forall \rho \in [0, 1/2))\,(\forall y \in \{\pm 1\})\,(\forall v \in \mathbb{R})\, \ell(y, v) = \alpha \cdot \bar{\ell}(y, v) + \beta.$$

By the definition of the noise-corrected loss (Equation 4), the given statement is that there exist $\alpha, \beta \colon [0, 1/2) \to \mathbb{R}$ with

$$(\forall \rho \in [0, 1/2))\,(\forall v \in \mathbb{R})\, \begin{bmatrix} \ell_1(v) \\ \ell_{-1}(v) \end{bmatrix} = \frac{\alpha(\rho)}{1 - 2\rho} \cdot \begin{bmatrix} 1 - \rho & -\rho \\ -\rho & 1 - \rho \end{bmatrix} \cdot \begin{bmatrix} \ell_1(v) \\ \ell_{-1}(v) \end{bmatrix} + \beta(\rho).$$

Adding together the two sets of equations (i.e. multiplying both sides by the all ones vector),

$$(\forall \rho \in [0, 1/2]) \, (\forall v \in \mathbb{R}) \, \ell_1(v) + \ell_{-1}(v) = \alpha(\rho) \cdot (\ell_1(v) + \ell_{-1}(v)) + 2\beta(\rho)$$
$$\iff (\forall \rho \in [0, 1/2]) \, (\forall v \in \mathbb{R}) \, (1 - \alpha(\rho)) \cdot (\ell_1(v) + \ell_{-1}(v)) = 2\beta(\rho)$$
$$\iff (\forall \rho \mid \alpha(\rho) \neq 1) \, (\forall v \in \mathbb{R}) \, \ell_1(v) + \ell_{-1}(v) = \frac{2\beta(\rho)}{1 - \alpha(\rho)},$$

which is a constant independent of $v$. If $\alpha(\rho) \equiv 1$, then clearly $\beta(\rho) \equiv 0$, and we have

$$(\forall v \in \mathbb{R}) \, (1 - \rho) \cdot \ell_1(v) - \rho \cdot \ell_{-1}(v) = (1 - 2\rho) \cdot \ell_1(v),$$

thus implying that $\ell_1(v) = \ell_{-1}(v)$. Such a loss is not interesting since it cannot possibly be classification calibrated. ☐

*Proof of Proposition 5.* ( $\impliedby$ ). Clearly for an $\ell$ satisfying the given condition, $\ell_1(v) + \ell_{-1}(v) = B + C$, a constant.

( $\implies$ ). By assumption, $\ell_1$ is convex. By the given condition, equivalently, $(\exists C \in \mathbb{R}) \, C - \ell_1$ is convex. But this is in turn equivalent to $-\ell_1$ also being convex. The only possibility for both $\ell_1$ and $-\ell_1$ being convex is that $\ell_1$ is affine, hence showing the desired implication. ☐

*Proof of Proposition 6.* Fix $\ell = \ell^{\mathrm{unh}}$. It is easy to check that

$$(\forall \eta \in [0, 1]) \, (\forall v \in \mathbb{R}) \, L_\ell(\eta, v) = (1 - 2\eta) \cdot v + 1, \tag{11}$$

and so

$$(\forall \eta \in [0, 1]) \, \underset{v \in [-B, +B]}{\mathrm{argmin}} \, L_\ell(\eta, v) = \begin{cases} +B & \text{if } \eta > \frac{1}{2} \\ -B & \text{else.} \end{cases}$$

It is not a coincidence that the above is a scaled version of the minimiser for the hinge loss. Trivially, for any $v \in [-B, +B]$, we have that

$$\begin{aligned} \ell(y, v) &= 1 - yv \\ &= B \cdot \left(1 - \frac{yv}{B}\right) + (1 - B) \\ &= B \cdot \max\left(0, 1 - \frac{yv}{B}\right) + (1 - B) \text{ as } yv \leq B \\ &= B \cdot \ell^{\mathrm{hinge}}\left(y, \frac{v}{B}\right) + (1 - B) \end{aligned} \tag{12}$$

where $\ell^{\mathrm{hinge}}(y, v) = \max(0, 1 - yv)$. It follows that

$$(\forall \eta \in [0, 1]) \, (\forall v \in [-B, +B]) \, L_\ell(\eta, v) = B \cdot L_{\mathrm{hinge}}\left(\eta, \frac{v}{B}\right) + (1 - B),$$

and so

$$\begin{aligned} (\forall \eta \in [0, 1]) \, \underset{v \in [-B, +B]}{\mathrm{argmin}} \, L_\ell(\eta, v) &= \underset{v \in [-B, +B]}{\mathrm{argmin}} \, L_{\mathrm{hinge}}\left(\eta, \frac{v}{B}\right) \\ &= B \cdot \underset{\tilde{v} \in [-1, +1]}{\mathrm{argmin}} \, L_{\mathrm{hinge}}(\eta, \tilde{v}) \\ &= B \cdot \begin{cases} +1 & \text{if } \eta > \frac{1}{2} \\ -1 & \text{else.} \end{cases} \end{aligned}$$

☐

*Proof of Proposition 7.* The basic idea of the surrogate regret bound can be seen visually. Figure 2 compares the zero-one, unhinged and hinge losses. If we restrict attention to $[-1, 1]$, the unhinged and hinge losses are identical, and surrogate regret bounds for the latter apply to the former. For general $B > 1$, we simply need to consider an appropriately scaled version of the hinge loss, and proceed identically.

Figure 2: Relationship between zero-one, unhinged and hinge loss.

Now we make this more formal. Fix $\ell = \ell^{\mathrm{unh}}$. Clearly, for any $v \in [-B, B]$,

$$\ell^{01}(y, v) \leq \ell^{\mathrm{hinge}}\left(y, \frac{v}{B}\right)$$

$$= \frac{1}{B} \cdot \ell(y, v) + 1 - \frac{1}{B} \text{ by Equation } 12.$$

Therefore, for any $s \in \mathcal{F}_B$,

$$\mathbb{L}_{01}^D(s) \leq \frac{1}{B} \cdot \mathbb{L}_\ell^D(s) + 1 - \frac{1}{B}.$$

Further, it is not hard to check that $\mathbb{L}_{01}^{D,\mathcal{F}_B,*} = \mathbb{L}_{\mathrm{hinge}}^{D,\mathcal{F}_B,*}$. Thus,

$$\mathbb{L}_{01}^{D,\mathcal{F}_B,*} = \mathbb{L}_{\mathrm{hinge}}^{D,\mathcal{F}_B,*}$$

$$= \mathop{\mathbb{E}}_{(\mathsf{X},\mathsf{Y})\sim D}\left[\ell^{\mathrm{hinge}}(\mathsf{Y}, \mathrm{sign}(2\eta(\mathsf{X}) - 1))\right]$$

$$= \mathop{\mathbb{E}}_{(\mathsf{X},\mathsf{Y})\sim D}\left[\frac{1}{B} \cdot \ell(\mathsf{Y}, B \cdot \mathrm{sign}(2\eta(\mathsf{X}) - 1)) + 1 - \frac{1}{B}\right] \text{ by Equation } 12$$

$$= \frac{1}{B} \cdot \mathbb{L}_\ell^{D,\mathcal{F}_B,*} + 1 - \frac{1}{B} \text{ by Proposition } 6.$$

Now, since the scorer $x \mapsto \mathrm{sign}(2\eta(x)-1) \in \mathcal{S}_{01}^{D,*} \cap \mathcal{F}_B$, we have that $\mathrm{regret}_{01}^{D,\mathcal{F}_B}(s) = \mathrm{regret}_{01}^D(s)$. Thus, for any $s \in \mathcal{F}_B$,

$$\mathrm{regret}_{01}^D(s) = \mathbb{L}_{01}^{D,\mathcal{F}_B}(s) - \mathbb{L}_{01}^{D,\mathcal{F}_B,*}$$

$$\leq \frac{1}{B} \cdot \left(\mathbb{L}_\ell^{D,\mathcal{F}_B}(s) - \mathbb{L}_\ell^{D,\mathcal{F}_B,*}\right)$$

$$= \frac{1}{B} \cdot \mathrm{regret}_\ell^D(s)$$

$$\leq \mathrm{regret}_\ell^D(s) \text{ since } B \geq 1.$$

To show the relation to the corrupted regret, by Equation 4, for $\ell = \ell^{\mathrm{unh}}$,

$$(\forall y \in \{\pm 1\})\,(\forall v \in \mathbb{R})\,\bar{\ell}(y, v) = \frac{1}{1 - 2\rho} \cdot \ell(y, v),$$

i.e. the unhinged loss is its own noise-corrected loss, with a scaling factor of $\frac{1}{1-2\rho}$. Thus, since the $\ell$-regret on $D$ and $\bar{\ell}$-regret on $\bar{D}$ coincide,

$$\mathrm{regret}_\ell^{D,\mathcal{F}_B}(s) = \mathrm{regret}_{\bar{\ell}}^{\bar{D},\mathcal{F}_B}(s) = \frac{1}{1 - 2\rho} \cdot \mathrm{regret}_\ell^{\bar{D},\mathcal{F}_B}(s).$$

$\square$

*Proof of Proposition 8.* While this has effectively been shown in Hastie et al. [2004, Section 6] by virtue of the connection between the unhinged loss and nearest centroid classification, we will prove this using a different technique. On a distribution $D$, a soft-margin SVM solves

$$\min_{w \in \mathcal{H}} \mathbb{E}_{(X,Y) \sim D} [\max(0, 1 - Y \cdot \langle w, \Phi(x) \rangle_{\mathcal{H}})] + \frac{\lambda}{2} \langle w, w \rangle_{\mathcal{H}}^2.$$

Let $w_{\text{hinge},\lambda}^*$ denote the optimal solution to this objective. Now, by Shalev-Shwartz et al. [2007, Theorem 1],

$$||w_{\text{hinge},\lambda}^*||_{\mathcal{H}} \leq \frac{1}{\sqrt{\lambda}}.$$

Now suppose that $R = \sup_{x \in \mathcal{X}} ||\Phi(x)||_{\mathcal{H}} < \infty$. Then, by the Cauchy-Schwartz inequality,

$$(\forall x \in \mathcal{X}) \, |\langle w_{\text{hinge},\lambda}^*, \Phi(x) \rangle_{\mathcal{H}}| \leq ||w_{\text{hinge},\lambda}^*||_{\mathcal{H}} \cdot ||\Phi(x)||_{\mathcal{H}} \leq \frac{R}{\sqrt{\lambda}}.$$

It follows that if $\lambda \geq R^2$, then

$$(\forall x \in \mathcal{X}) \, |\langle w_{\text{hinge},\lambda}^*, \Phi(x) \rangle_{\mathcal{H}}| \leq 1.$$

But this means that we never activate the flat portion of the hinge loss. Thus, for $\lambda \geq R^2$, the SVM objective is equivalent to

$$\min_{w \in \mathcal{H}} \mathbb{E}_{(X,Y) \sim D} [1 - Y \cdot \langle w, \Phi(x) \rangle_{\mathcal{H}}] + \frac{\lambda}{2} \langle w, w \rangle_{\mathcal{H}}^2.$$

which means the optimal solution will coincide with that of the regularised unhinged loss. Therefore, we can view unhinged loss minimisation as corresponding to learning a highly regularised SVM[7].

$\square$

*Proof of Proposition 9.* Fix some distribution $D$. Let

$$\mu = \mathbb{E}_{(X,Y) \sim D} [Y \cdot \Phi(X)]$$

be the optimal unhinged solution with regularisation strength $\lambda = 1$. Observe that $||\mu||_{\mathcal{H}} \leq R = \sup_{x \in \mathcal{X}} ||\Phi(x)||_{\mathcal{H}} < \infty$. For some $r > 0$, let

$$w_\phi^* = \operatorname*{argmin}_{||w||_{\mathcal{H}} \leq r} \mathbb{L}_\phi^D(w)$$

be the optimal $\phi$ solution with norm bounded by $r$. Similarly, let

$$w_{\text{unh}}^* = ||w_\phi^*|| \cdot \frac{\mu}{||\mu||_{\mathcal{H}}}$$

be the optimal unhinged solution with the same norm as the optimal $\phi$ solution. We will show that these two vectors have similar unhinged risks, and use this to show that the corresponding unit vectors must be close.

By definition, a convex potential has $\phi'(0) < 0$. As scaling of a loss does not affect its optimal solution, without loss of generality, we can assume $\phi'(0) = -1$. Then, since $\phi$ is convex, it is lower bounded by the linear approximation at zero:

$$(\forall v \in \mathbb{R}) \, \phi(v) + 1 - \phi(0) \geq 1 - v.$$

Observe that the RHS is the unhinged loss. Thus, the unhinged risk of a candidate solution can be bounded by its $\phi$ counterpart. Now we compare the unhinged and $\phi$ optimal solutions in terms of their unhinged risks:

$$\begin{aligned}
\mathbb{L}_{\text{unh}}^D(w_\phi^*) - \mathbb{L}_{\text{unh}}^D(w_{\text{unh}}^*) &\leq \mathbb{L}_\phi(w_\phi^*) - \mathbb{L}_{\text{unh}}(w_{\text{unh}}^*) + 1 - \phi(0) \\
&\leq \mathbb{L}_\phi^D(w_{\text{unh}}^*) - \mathbb{L}_{\text{unh}}^D(w_{\text{unh}}^*) + 1 - \phi(0) \text{ by optimality of } w_\phi^* \\
&= \mathbb{E}_{(X,Y) \sim D} \left[ \tilde{\phi}(Y \langle w_{\text{unh}}^*, \Phi(X) \rangle_{\mathcal{H}}) \right],
\end{aligned}$$

(13)

where $\tilde{\phi}\colon v \mapsto \phi(v) - \phi(0) + v$, the remainder term from the linear approximation to $\phi$. By Cauchy-Schwartz, we can restrict attention in Equation 13 to the behaviour of $\tilde{\phi}$ in the interval

$$I = [-||w_{\text{unh}}^*||_{\mathcal{H}} \cdot R, ||w_{\text{unh}}^*||_{\mathcal{H}} \cdot R],$$

where $R = \sup_{x \in \mathcal{X}} ||\Phi(x)||_{\mathcal{H}} < \infty$.

Now, by Taylor's remainder theorem,

$$(\forall v \in (-1,1))\ \tilde{\phi}(v) \leq \frac{a}{2}v^2, \tag{14}$$

where $a = \max_{v \in [-1,1]} \phi''(v) < +\infty$. Therefore, if $r \leq \frac{1}{R}$, $I \subseteq [-1,1]$ and so

$$\mathbb{L}_{\text{unh}}^D(w_\phi^*) - \mathbb{L}_{\text{unh}}^D(w_{\text{unh}}^*) \leq \frac{a}{2} \cdot \mathop{\mathbb{E}}_{(\mathsf{X},\mathsf{Y}) \sim D} \left[ \langle w_{\text{unh}}^*, \Phi(\mathsf{X}) \rangle_{\mathcal{H}}^2 \right] \text{ by Equation 14}$$

$$\leq \frac{a}{2} \cdot \mathop{\mathbb{E}}_{\mathsf{X} \sim M} \left[ ||w_{\text{unh}}^*||_{\mathcal{H}}^2 \cdot ||\Phi(\mathsf{X})||_{\mathcal{H}}^2 \right] \text{ by Cauchy-Schwartz}$$

$$\leq \frac{aR^2}{2} \cdot ||w_{\text{unh}}^*||_{\mathcal{H}}^2.$$

Now, the unhinged risk is

$$\mathbb{L}_{\text{unh}}^D(w) = 1 - \langle w, \mu \rangle_{\mathcal{H}}.$$

Thus,

$$-\langle w_\phi^*, \mu \rangle_{\mathcal{H}} + \langle w_{\text{unh}}^*, \mu \rangle_{\mathcal{H}} \leq \frac{aR^2}{2} \cdot ||w_{\text{unh}}^*||_{\mathcal{H}}^2.$$

Rearranging the above,

$$\langle w_\phi^*, \mu \rangle_{\mathcal{H}} \geq \langle w_{\text{unh}}^*, \mu \rangle_{\mathcal{H}} - \frac{aR^2}{2} \cdot ||w_{\text{unh}}^*||_{\mathcal{H}}^2$$

$$= ||w_\phi^*||_{\mathcal{H}} \cdot ||\mu||_{\mathcal{H}} - \frac{aR^2}{2} \cdot ||w_\phi^*||_{\mathcal{H}}^2 \text{ by definition of } w_{\text{unh}}^*$$

$$= ||w_\phi^*||_{\mathcal{H}} \cdot ||\mu||_{\mathcal{H}} \cdot \left(1 - \frac{aR^2}{2||\mu||_{\mathcal{H}}} \cdot ||w_\phi^*||_{\mathcal{H}}\right)$$

$$\geq ||w_\phi^*||_{\mathcal{H}} \cdot ||\mu||_{\mathcal{H}} \cdot \left(1 - \frac{aR^2}{2||\mu||_{\mathcal{H}}} \cdot r\right) \text{ since } ||w_\phi^*||_{\mathcal{H}} \leq r.$$

Thus, for $\epsilon = \frac{aR^2}{2||\mu||_{\mathcal{H}}}$,

$$\left\langle \frac{w_\phi^*}{||w_\phi^*||_{\mathcal{H}}}, \frac{\mu}{||\mu||_{\mathcal{H}}} \right\rangle_{\mathcal{H}} \geq 1 - \epsilon.$$

It follows that the two unit vectors can be made arbitrarily close to each other by decreasing $r$. Since this corresponds to increasing the strength of regularisation (by Lagrange duality), and since $\frac{\mu}{||\mu||_{\mathcal{H}}}$ corresponds to the normalised unhinged solution for any regularisation strength, we may conclude that

$$(\forall \epsilon > 0)\,(\exists \lambda_0 > 0)\,(\forall \lambda > \lambda_0)\ \left|\left| \frac{w_{\phi,\lambda}^*}{||w_{\phi,\lambda}^*||_{\mathcal{H}}} - \frac{w_{\text{unh},\lambda}^*}{||w_{\text{unh},\lambda}^*||_{\mathcal{H}}} \right|\right|_{\mathcal{H}}^2 \leq \epsilon,$$

and the result follows. $\qquad\square$

## A.1 Additional helper lemmas

**Lemma 10.** *A pair of losses $(\ell, \tilde{\ell})$ are order equivalent if and only if*

$$(\exists \alpha \in R_+, \beta \in \mathbb{R})\ \tilde{\ell}(y,v) = \alpha \cdot \ell(y,v) + \beta.$$

*Proof of Lemma 10.* Recall that order equivalence of $(\ell, \tilde{\ell})$ is (Definition 4)

$$(\forall R, R') \mathop{\mathbb{E}}_{(\mathsf{Y},\mathsf{S})\sim R}[\ell(\mathsf{Y},\mathsf{S})] \leq \mathop{\mathbb{E}}_{(\mathsf{Y},\mathsf{S})\sim R'}[\ell(\mathsf{Y},\mathsf{S})] \iff \mathop{\mathbb{E}}_{(\mathsf{Y},\mathsf{S})\sim R}\left[\tilde{\ell}(\mathsf{Y},\mathsf{S})\right] \leq \mathop{\mathbb{E}}_{(\mathsf{Y},\mathsf{S})\sim R'}\left[\tilde{\ell}(\mathsf{Y},\mathsf{S})\right].$$

Now define the utility functions
$$U \colon (y, s) \mapsto -\ell(y, s)$$
and
$$V \colon (y, s) \mapsto -\tilde{\ell}(y, s).$$

Then, order equivalence is trivially equivalent to

$$(\forall R, R') \mathop{\mathbb{E}}_{(\mathsf{Y},\mathsf{S})\sim R}[U(\mathsf{Y},\mathsf{S})] \geq \mathop{\mathbb{E}}_{(\mathsf{Y},\mathsf{S})\sim R'}[U(\mathsf{Y},\mathsf{S})] \iff \mathop{\mathbb{E}}_{(\mathsf{Y},\mathsf{S})\sim R}[V(\mathsf{Y},\mathsf{S})] \geq \mathop{\mathbb{E}}_{(\mathsf{Y},\mathsf{S})\sim R'}[V(\mathsf{Y},\mathsf{S})].$$

That is, the utility functions $U, V$ specify the same ordering over distributions. By DeGroot [1970, Section 7.9, Theorem 2], this is possible if and only if $U, V$ are affinely related:

$$(\exists \alpha \in \mathbb{R}_+, \beta \in \mathbb{R})\,(\forall y, s)\, U(y, s) = \alpha \cdot V(y, s) + \beta.$$

The result follows by re-expressing this in terms of $\ell$ and $\tilde{\ell}$. □

# Additional Discussion for "Learning with Symmetric Label Noise: The Importance of Being Unhinged"

## B  Evidence that non-convex losses and linear scorers may not be SLN-robust

We now present preliminary evidence that for $\ell$ being the TangentBoost loss,

$$\ell(y, v) = (2\tan^{-1}(yv) - 1)^2,$$

or the $t$-logistic regression loss for $t = 2$,

$$\ell(y, v) = \log(1 - yv + \sqrt{1 + v^2}),$$

$(\ell, \mathcal{F}_{\text{lin}})$ may not be SLN-robust. We do this by looking at the minimisers of these losses on the 2D example of Long and Servedio [2010]. Of course, as these losses are non-convex, exact minimisation of the risk is challenging. However, as the search space is $\mathbb{R}^2$, we construct a grid of resolution $0.025$ over $[-10, 10]^2$. We then exhaustively compute the objective for all grid points, and seek the minimiser.

We apply this procedure to the Long and Servedio [2010] dataset with $\gamma = \frac{1}{60}$, and with a 30% noise rate. Figure 3 plots the results of the objective for the TangentBoost loss. We find that the minimiser is at $w^* = (0.2, 1.3)$. This results in a classifier with error rate of $\frac{1}{2}$ on $D$. Similarly, from Figure 4, we find that the minimiser is $w^* = (1.025, 5.1)$, which also results in a classifier with error rate of $\frac{1}{2}$.

Figure 3: Risk values for various weight vectors $w = (w_1, w_2)$, TangentBoost, Long and Servedio [2010] dataset.

The shape of these plots suggests that the minimiser is indeed found in the interval $[-10, 10]^2$. To further verify this, we performed L-BFGS minimisation of these losses using 100 different random initialisations, uniformly from $[-100, 100]^2$. We find that in each trial, the Tangent-Boost solution converges to $w^* = (0.2122, 1.3031)$, while the $t$-logistic solution converges to $w^* = (1.0372, 5.0873)$, both of which result in accuracy of 50% on $D$.

### B.1  Conjecture: (most) strictly proper composite losses are not SLN-robust

Recall that a loss $\ell$ is *strictly proper composite* [Reid and Williamson, 2010] if its (unique) Bayes-optimal scorer is some strictly monotone transformation $\psi$ of the class-probability function:

Figure 4: Risk values for various weight vectors $w = (w_1, w_2)$, $t$-logistic regression, Long and Servedio [2010] dataset.

$(\forall D)\, \mathcal{S}_\ell^{D,*} = \{\psi \circ \eta\}$. It is easy to check that both the TangentBoost and $t$-logistic losses are proper composite. We conjecture the above is a manifestation of the following phenomenon.

**Conjecture 1.** *Pick any strictly proper composite (but not necessarily convex) $\ell$ whose link function has range $\mathbb{R}$. Then, $(\ell, \mathcal{F}_{\mathrm{lin}})$ is not SLN-robust.*

We believe the above is true for the following reason. Suppose $D$ is some linearly separable distribution, with $\eta\colon x \mapsto [\![\langle w^*, x\rangle > 0]\!]$ for some $w^*$. Then, minimising $\ell$ with $\mathcal{F}_{\mathrm{lin}}$ will be well-specified: the Bayes-optimal scorer is $\psi([\![\langle w^*, x\rangle > 0]\!])$. If the range of $\psi$ is $\mathbb{R}$, then this is equivalent to $\infty \cdot (2[\![\langle w^*, x\rangle > 0]\!] - 1)$, which is in $\mathcal{F}_{\mathrm{lin}}$ if we allow for the extended reals. The resulting classifier will thus have $100\%$ accuracy. However, by injecting any non-zero label noise, minimising $\ell$ with $\mathcal{F}_{\mathrm{lin}}$ will no longer be well-specified, as $\bar{\eta}$ takes on the values $\{1 - \rho, \rho\}$, which cannot be the sole set of output scores for any linear scorer if $|\mathcal{X}| > 3$. We believe it unlikely that every such misspecified solution have $100\%$ accuracy on $D$. We further believe it likely that one can exhibit a scenario, possibly the same as the Long and Servedio [2010] example, where the resulting solution has accuracy $50\%$.

Two further comments are in order. First, if a loss is strictly proper composite, then it cannot satisfy Equation 5, and hence it cannot be strongly SLN-robust. (However, this does leave open the possibility that with $\mathcal{F}_{\mathrm{lin}}$, the loss is SLN-robust.) Second, observe that the restriction that $\psi$ have range $\mathbb{R}$ is necessary to rule out cases such as square loss, where the link function has range $[-1, 1]$.

### B.2 In defence of non-convex losses: beyond SLN-robustness

The above illustrates the possible non SLN-robustness of two non-convex losses. However, there may be *other* notions under which these losses are robust. For example, Ding and Vishwanathan [2010] defines robustness to be a stability of the asymptotic maximum likelihood solution when adding a new labelled instance (chosen *arbitrarily* from $\mathcal{X} \times \{\pm 1\}$), based on a definition in O'Hagan [1979]. Intuitively, this captures robustness to outliers in the instance space, so that e.g. an adversarial mislabelling of an instance far from the true decision boundary does not adversely affect the learned model. Such a notion of robustness is clearly of practical interest, and future study of such alternate notions would be of value. (Appendix D.2 highlights some guarantees that are possible instance dependent noise, but still in the regime where instances are drawn from the true marginal $M$.)

## C  Preservation of mean maps

Pick any $D$, and $\rho \in [0, 1/2)$. Then,

$$(\forall x \in \mathfrak{X})\, 2\bar{\eta}(x) - 1 = 2 \cdot ((1 - 2\rho) \cdot \eta(x) + \rho) - 1$$
$$= (1 - 2\rho) \cdot (2\eta(x) - 1).$$

Thus, for any feature mapping $\Phi \colon \mathfrak{X} \to \mathcal{H}$, the kernel mean map of the clean distribution is

$$\mathbb{E}_{(\mathsf{X},\mathsf{Y}) \sim D} [\mathsf{Y} \cdot \Phi(\mathsf{X})] = \mathbb{E}_{\mathsf{X} \sim M} [(2\eta(\mathsf{X}) - 1) \cdot \Phi(\mathsf{X})]$$

$$= \frac{1}{(1 - 2\rho)} \cdot \mathbb{E}_{\mathsf{X} \sim M} [(2\bar{\eta}(\mathsf{X}) - 1) \cdot \Phi(\mathsf{X})]$$

$$= \frac{1}{(1 - 2\rho)} \cdot \mathbb{E}_{(\mathsf{X},\mathsf{Y}) \sim \mathrm{SLN}(D,\rho)} [\mathsf{Y} \cdot \Phi(\mathsf{X})],$$

which is a scaled version of the kernel mean map of the noisy distribution. That is, the kernel mean map is preserved under symmetric label noise. Instantiating the above with a specific $x \in \mathfrak{X}$ gives Equation 9.

## D  Additional theoretical considerations

We discuss some further theoretical properties of the unhinged loss.

### D.1  Generalisation bounds

Generalisation bounds are readily derived for the unhinged loss. For a training sample $\mathsf{S} \sim D^n$, define the $\ell$-deviation of a scorer $s \colon \mathfrak{X} \to \mathbb{R}$ to be the difference in its population and empirical $\ell$-risk,

$$\mathrm{dev}_\ell^{D,\mathsf{S}}(s) = \mathbb{L}_\ell^D(s) - \mathbb{L}_\ell^{\mathsf{S}}(s).$$

This quantity is of interest because a standard result says that for the empirical risk minimiser $s_n$ over some function class $\mathcal{F}$, $\mathrm{regret}_\ell^{D,\mathcal{F}}(s_n) \leq 2 \cdot \sup_{s \in \mathcal{F}} |\mathrm{dev}_\ell^{D,\mathsf{S}}(s)|$ [Boucheron et al., 2005, Equation 2]. For unhinged loss, we have the following Rademacher based bound.

**Proposition 11.** *Pick any $D$ and $n \in \mathbb{N}_+$. Let $\mathsf{S} \sim D^n$ denote an empirical sample. For some $B \in \mathbb{R}_+$, let $s \in \mathcal{F}_B$. Then, with probability at least $1 - \delta$ over the choice of $\mathsf{S}$, for $\ell = \ell^{\mathrm{unh}}$,*

$$\mathrm{dev}_\ell^{D,\mathsf{S}}(s) \leq 2 \cdot \mathcal{R}_n(\mathcal{F}_B, \mathsf{S}) + B \cdot \sqrt{\frac{\log \frac{2}{\delta}}{2n}}$$

*where $\mathcal{R}_n(\mathcal{F}_B, \mathsf{S})$ is the empirical Rademacher complexity of $\mathcal{F}_B$ on sample $\mathsf{S}$.*

*Proof of Proposition 11.* The standard Rademacher-complexity generalisation bound [Bartlett and Mendelson, 2002, Theorem 7], [Boucheron et al., 2005, Theorem 4.1] states that with probability at least $1 - \delta$ over the choice of $\mathsf{S}$,

$$\mathrm{dev}_\ell^{D,\mathsf{S}}(s) \leq 2 \cdot ||(\ell)'||_\infty \cdot \mathcal{R}_n(\mathcal{F}_B, \mathsf{S}) + ||\ell||_\infty \cdot \sqrt{\frac{\log \frac{2}{\delta}}{2n}}.$$

For the unhinged loss, $||(\ell^{\mathrm{unh}})'||_\infty = 1$. Further, since we work over bounded scorers, $||\ell^{\mathrm{unh}}||_\infty = B$. The result follows. $\qquad\square$

Proposition 11 holds equally when learning from a corrupted sample $\bar{\mathsf{S}} \sim \bar{D}^n$. Since $\mathrm{regret}_{\ell^{\mathrm{unh}}}^{D,\mathcal{F}}(s_n) = \frac{1}{1-2\rho} \cdot \mathrm{regret}_{\ell^{\mathrm{unh}}}^{\bar{D},\mathcal{F}}(s_n)$ by Proposition 7, by minimising the unhinged loss on the corrupted sample, we can bound the regret on the clean distribution.

### D.2 The unhinged loss and instance dependent noise

The SLN model is a special case of the following (more realistic) *non uniform* noise model: given a notional clean distribution $D$, one observes samples from a corrupted distribution $\bar{D}$, where labels are flipped with some *instance dependent* probability $\rho(x) \in [0, 1/2)$. It is not hard to check that the unhinged solution will no longer be perfectly robust under this noise model. However, Servedio [1999] showed that if one further assumes $M$ is uniform over the unit sphere, a nearest centroid classifier will be robust. More generally, one might hope that one can at least *bound* the degradation in performance under the noise model. Ghosh et al. [2015, Remark 1] showed that this is indeed possible for losses $\ell$ satisfying Equation 5: a simple argument reveals a bound on the $\ell$-risk of the minimiser on the *corrupted* distribution in terms of the risk of the minimiser on the *clean* distribution, and a scaling factor that depends on the highest noise rate over all instances,

$$\mathbb{L}_\ell^D(S_\ell^{\bar{D},\mathcal{F},*}) = \frac{\mathbb{L}_\ell^D(S_\ell^{D,\mathcal{F},*})}{1 - 2\max_{x \in \mathcal{X}} \rho(x)}.$$

As the unhinged loss satisfies Equation 5, this immediately implies a guarantee for the more realistic instance dependent noise case. Further study of this subject will be the matter of future work.

### D.3 On balanced error and area under the ROC immunity

Menon et al. [2015] recently showed that with rich function classes, class-probability estimation techniques will be robust to a general class of corruptions in terms of ranking performance (as measured by the area under the ROC curve or AUC) as well as classification performance (as measured by the balanced error or BER). In particular, they showed an affine relationship between the BER for an arbitrary classifier on the clean and corrupted distributions, and similarly for the AUC. This means that minimisers of the BER on the clean and corrupted distributions coincide for *any* function class $\mathcal{F}$; however, this does not mean that the minimisers for a *surrogate* to the BER coincide on the clean and corrupted distributions. In order to establish that surrogate minimisation is sensible for the BER, Menon et al. [2015] rely on the choice of $\mathcal{F} = \mathbb{R}^{\mathcal{X}}$. When $\mathcal{F} = \mathcal{F}_{\text{lin}}$, the example of Long and Servedio [2010] shows that a class-probability estimation technique such as logistic regression may perform poorly. By contrast, we emphasise that the unhinged loss' robustness holds as-is for any choice of $\mathcal{F}$.

## E  Additional relations to existing methods

We discuss some further connections of the unhinged loss to existing methods.

### E.1 Unhinging the SVM

We can motivate the unhinged loss intuitively by studying the noise-corrected versions of the hinge loss, as per Equation 4. Figure 5 shows the noise corrected hinge loss for $\rho \in \{0, 0.2, 0.4\}$. We see that as the noise rate increases, the effect is to slightly *unhinge* the original loss, by removing its flat portion[8]. Thus, if we knew the noise rate $\rho$, we could use these *slightly unhinged* losses to learn.

Of course, in general we do not know the noise rate. Further, the slightly unhinged losses are non-convex. So, in order to be robust to an *arbitrary* noise rate $\rho$, we can *completely unhinge* the loss, yielding

$$\ell_1^{\text{unh}}(v) = 1 - v \text{ and } \ell_{-1}^{\text{unh}}(v) = 1 + v.$$

### E.2 Relation to centroid classifiers

As established in §6.1, the optimal unhinged classifier (Equation 8) is equivalent to a centroid classifier, where one replaces the positive and negative classes by their centroids, and performs classification based on the distance of an instance to the two centroids. Such a classifier has been proposed

Figure 5: Noise-corrected versions of hinge loss, $\ell_1(v) = \max(0, 1 - v)$. Best viewed in colour.

as a prototypical example of a simple kernel-based classifier [Schölkopf and Smola, 2002, Section 1.2], [Shawe-Taylor and Cristianini, 2004, Section 5.1] Balcan et al. [2008, Definition 4] considers such classification rules using general similarity functions in place of kernels corresponding to an RKHS.

The optimal unhinged classifier is also closely related to the Rocchio classifier in information retrieval [Manning et al., 2008, pg. 181], and the nearest centroid classifier in computational genomics [Tibshirani et al., 2002]. The optimal kernelised scorer for these approaches is [Doloc-Mihu et al., 2003]

$$ s^* \colon x \mapsto \left( \mathop{\mathbb{E}}_{\mathsf{X} \sim P} \left[ k(\mathsf{X}, x) \right] - \mathop{\mathbb{E}}_{\mathsf{X} \sim Q} \left[ k(\mathsf{X}, x) \right] \right), $$

i.e. it does not weight each of the kernel means.

### E.3   Relation to kernel density estimation

When working with an RKHS with a translation invariant kernel[9], the optimal unhinged scorer (Equation 8) can be interpreted as follows: perform kernel density estimation on the positive and negative classes, and then classify instances according to Bayes' rule. For example, with a Gaussian RBF kernel, the classifier is equivalent to using a Gaussian kernel to compute density estimates of $P, Q$, and using these to classify. This is known as a kernel classification rule [Devroye et al., 1996, Chapter 10].

This perspective suggests that in computing $s^*_{\mathrm{unh}, \lambda}$, we may also estimate the corrupted class-probability function. In particular, observe that if we compute $\frac{\pi}{1-\pi} \cdot \frac{\mathbb{E}_{\mathsf{X} \sim P}[k(\mathsf{X}, x)]}{\mathbb{E}_{\mathsf{X} \sim Q}[k(\mathsf{X}, x)]}$, similar to the Nadaraya-Watson estimator [Bishop, 2006, pg. 300], then this provides an estimate of $\frac{\eta(x)}{1-\eta(x)}$. Of course, such an approach will succumb to the curse of dimensionality[10].

An alternative is to use the Probing reduction [Langford and Zadrozny, 2005], by computing an ensemble of cost-sensitive classifiers at varying cost ratios. To this end, observe that the following weighted unhinged (or *whinge*) loss,

$$ \ell_1^{\mathrm{whinge}}(v) = c_1 \cdot -v $$
$$ \ell_{-1}^{\mathrm{whinge}}(v) = c_{-1} \cdot v $$

for some $c_{-1} \in [0, 1]$ and $c_1 = 1 - c_{-1}$, will have a restricted Bayes-optimal scorer of $B \cdot \mathrm{sign}(\eta(x) - c_{-1})$ over $\mathcal{F}_B$. Further, it will result in an optimal scorer that simply weights each of the kernel

means,

$$s^*_{\text{whinge},\lambda} \colon x \mapsto \frac{1}{\lambda} \cdot \mathop{\mathbb{E}}_{(\mathsf{X},\mathsf{Y}) \sim D} \left[ c_\mathsf{Y} \cdot \mathsf{Y} \cdot k(\mathsf{X}, x) \right],$$

making it trivial to compute as $c$ is varied.

### E.4 Relation to the MMD witness

The optimal weight vector for unhinged loss (Equation 7) can be expressed as

$$w^*_{\text{unh},\lambda} = \frac{1}{\lambda} \cdot (\pi \cdot \mu_P - (1 - \pi) \cdot \mu_Q),$$

where $\mu_P$ and $\mu_Q$ are the *kernel mean maps* with respect to $\mathcal{H}$ of the positive and negative class-conditionals distributions,

$$\mu_P = \mathop{\mathbb{E}}_{\mathsf{X} \sim P} [\Phi(\mathsf{X})]$$
$$\mu_Q = \mathop{\mathbb{E}}_{\mathsf{X} \sim Q} [\Phi(\mathsf{X})].$$

When $\pi = \frac{1}{2}$, $\|w_1^*\|_\mathcal{H}$ is precisely the *maximum mean discrepancy* (*MMD*) [Gretton et al., 2012] between $P$ and $Q$, using all functions in the unit ball of $\mathcal{H}$. The mapping $x \mapsto \langle w_1^*, x \rangle_\mathcal{H}$ itself is referred to as the *witness function* [Gretton et al., 2012, §2.3]. While the motivation of MMD is to perform hypothesis testing so as to distinguish between two distributions $P, Q$, rather than constructing a suitable scorer, the fact that it arises from the optimal scorer for the unhinged loss has been previously noted [Sriperumbudur et al., 2009, Theorem 1].

## F  Example of poor classification with square loss

We illustrate that square loss with a linear function class may perform poorly even when the underlying distribution is linearly separable. We consider the dataset of Long and Servedio [2010], with *no* label noise. That is, we have $\mathcal{X} = \{(1, 0), (\gamma, 5\gamma), (\gamma, -\gamma), (\gamma, -\gamma)\} \subset \mathbb{R}^2$, and $\eta \colon x \mapsto 1$. Let $X \in \mathbb{R}^{4 \times 2}$ be the feature matrix of the four data points. Then, the optimal weight vector for square loss is

$$w^* = (X^T X)^{-1} X^T \begin{bmatrix} 1 \\ 1 \\ 1 \\ 1 \end{bmatrix}$$
$$= \begin{bmatrix} \frac{8\gamma + 3}{8\gamma^2 + 3} \\ -\frac{\gamma + 1}{3\gamma \cdot (8\gamma^2 + 3)} \end{bmatrix}.$$

It is easy to check that the predicted scores are then

$$s^* = \begin{bmatrix} \frac{8\gamma + 3}{8\gamma^2 + 3} \\ \frac{\gamma \cdot (8\gamma + 3)}{8\gamma^2 + 3 - \frac{5 \cdot (\gamma - 1)\gamma}{24\gamma^3 + 9\gamma}} \\ \frac{(\gamma - 1) \cdot \gamma}{24\gamma^3 + 9\gamma + \frac{\gamma \cdot (8\gamma + 3)}{8\gamma^2 + 3}} \\ \frac{(\gamma - 1) \cdot \gamma}{24\gamma^3 + 9\gamma + \frac{\gamma \cdot (8\gamma + 3)}{8\gamma^2 + 3}} \end{bmatrix}.$$

But for $\gamma < \frac{1}{12}$, this means that the predicted scores for the last two examples are negative. That is, the resulting classifier will have 50% accuracy. (This does not contradict the robustness of square loss, as robustness simply requires that performance is the *same* with and without noise.)

It is initially surprising that square loss fails in this example, as we are employing a linear function class, and the true $\eta$ is expressible as a linear function. However, recall that the Bayes-optimal scorer for square loss is

$$\mathcal{S}^{D,*}_\ell = \{s \colon x \mapsto 2\eta(x) - 1\}.$$

In this case, the Bayes-optimal scorer is

$$s^* \colon x \mapsto 2[\![x_1 > 0]\!] - 1.$$

The application of a threshold means the that scorer is *not expressible as a linear model*. Therefore, the combination of loss and function class is in fact *not* well-specified for the problem. By contrast, consider the use of the squared hinge loss, $\ell(y, v) = \max(0, 1 - yv)^2$. This loss induces a *set* of Bayes-optimal scorers, which are:

$$\mathcal{S}_\ell^{D,*} = \left\{ s \mid (\forall x \in \mathcal{X}) \begin{cases} \eta(x) = 1 & \implies s(x) \in [1, \infty) \\ \eta(x) \in (0, 1) & \implies s(x) = 2\eta(x) - 1 \\ \eta(x) = 0 & \implies s(x) \in (-\infty, 1]. \end{cases} \right\}$$

Crucially, we *can* find a linear scorer that is in this set: for, say, $v = (\frac{1}{\gamma}, 0)$, we clearly have $\langle v, x \rangle \geq 1$ for every $x \in \mathcal{X}$, and so this is a Bayes-optimal scorer. Thus, minimising the square hinge loss on this distribution will indeed find a classifier with $100\%$ accuracy.

## G    Example of poor classification with unhinged loss

We illustrate that the unhinged loss with a linear function class may perform poorly even when the underlying distribution is linearly separable. (For another example where instances are on the unit ball, see Balcan et al. [2008, Figure 1].) Consider a distribution $D_{M,\eta}$ uniformly concentrated on $\mathcal{X} = \{x_1, x_2, x_3\}$ with $x_1 = (1, 2), x_2 = (1, -4), x_3 = (-1, 1)$, with $\eta(x_1) = \eta(x_2) = 1$ and $\eta(x_3) = 0$, i.e. the first two instances are positive, and the third instance negative. Then it is evident that the optimal unhinged hyperplane, with regularisation strength 1, is $w^* = (1, -1)$. This will misclassify the first instance as being negative. Figure 6 illustrates.

It is easy to check that for this particular distribution, the optimal weight for square loss is $w^* = (1, 0)$. This results in perfect classification. Thus, we have a reversal of the scenario of the previous section – here, square loss classifies perfectly, while the unhinged loss classifies no better than random guessing.

It may appear that the above contradicts the classification-calibration of the unhinged loss: there certainly is a linear scorer that is Bayes-optimal over $\mathcal{F}_B$, namely, $w^* = (B, 0)$. The subtlety is that in this case, minimisation over the unit ball $||w||_2 \leq 1$ (as implied by $\ell_2$ regularisation) is unable to restrict attention to the desired scorer.

There are two ways to rectify examples such as the above. First, as in general, we can employ a suitably rich kernel, e.g. a Gaussian RBF kernel. It is not hard to verify that on this dataset, such a kernel will find a perfect classifier. Second, we can look to explicitly enforce that minimisation is over all $w$ satisfying $|\langle w, x_n \rangle| \leq 1$. This will result in a linear program (LP) that may be solved easily, but does not admit a closed form solution as in the case of minimising over the unit ball. It may be checked that the resulting LP will recover the optimal weight $w^* = (1, 0)$. While this approach is suitable for this particular example, issues arise when dealing with infinite dimensional feature mappings (as we lose the existence of a representer theorem without regularisation based on the norm in the Hilbert space [Yu et al., 2013]).

Figure 6: Example of linearly separable distribution where, when learning with the unhinged loss and a linear function class, the resulting hyperplane (in red) misclassifies one of the instances.

# Additional Experiments for "Learning with Symmetric Label Noise: The Importance of Being Unhinged"

## H  Additional experimental results

Table 4 reports the 0-1 error for a range of losses on the Long and Servedio [2010] dataset. TanBoost refers to the loss of Masnadi-Shirazi et al. [2010]. As before, we find the unhinged loss to generally find a good classifier. Observe that the relatively poor performance of the square and TanBoost loss can be attributed to the findings of Appendix B, F.

We next report the 0-1 error and one minus the AUC for a range of datasets. We begin with a dataset of Mease and Wyner [2008], where $\mathcal{X} = [0,1]^{20}$, and $M$ is the uniform distribution. Further, we have $\eta\colon x \mapsto [\![\langle w^*, x\rangle > 2.5]\!]$ for $w^* = [\mathbf{1}_5 \quad \mathbf{0}_{15}]$, i.e. there is a sparse separating hyperplane. Table 5 reports the results on this dataset injected with various levels of symmetric noise. On this dataset, the $t$-logistic loss generally performs the best.

Finally, we report the 0-1 error and one minus the AUC on some UCI datasets in Tables 6 – 7. Table 3 summarises statistics of the UCI data. Several datasets are imbalanced, meaning that 0-1 error is not the ideal measure of performance (as it can be made small with a trivial majority classifier). The AUC is thus arguably a better indication of performance for these datasets. We generally find that at high noise rates (40%), the AUC of the unhinged loss is superior to that of other losses.

| Dataset | $N$ | $D$ | $\mathbb{P}(\mathsf{Y} = 1)$ |
|---------|-----|-----|------------------------------|
| Iris | 150 | 4 | 0.3333 |
| Ionosphere | 351 | 34 | 0.3590 |
| Housing | 506 | 13 | 0.0692 |
| Car | 1,728 | 8 | 0.0376 |
| USPS 0v7 | 2,200 | 256 | 0.5000 |
| Splice | 3,190 | 61 | 0.2404 |
| Spambase | 4,601 | 57 | 0.3940 |

Table 3: Summary of UCI datasets. Here, $N$ denotes the total number of samples, and $D$ the dimensionality of the feature space.

|  | **Hinge** | **Logistic** | **Square** | $t$-**logistic** | **TanBoost** | **Unhinged** |
|---|---|---|---|---|---|---|
| $\rho = 0$ | $0.00 \pm 0.00$ | $0.00 \pm 0.00$ | $0.25 \pm 0.00$ | $0.00 \pm 0.00$ | $0.25 \pm 0.00$ | $0.00 \pm 0.00$ |
| $\rho = 0.1$ | $0.15 \pm 0.27$ | $0.24 \pm 0.05$ | $0.25 \pm 0.00$ | $0.00 \pm 0.00$ | $0.25 \pm 0.00$ | $0.00 \pm 0.00$ |
| $\rho = 0.2$ | $0.21 \pm 0.30$ | $0.25 \pm 0.00$ | $0.25 \pm 0.00$ | $0.00 \pm 0.00$ | $0.25 \pm 0.00$ | $0.00 \pm 0.00$ |
| $\rho = 0.3$ | $0.38 \pm 0.37$ | $0.25 \pm 0.03$ | $0.25 \pm 0.02$ | $0.22 \pm 0.08$ | $0.25 \pm 0.03$ | $0.00 \pm 0.00$ |
| $\rho = 0.4$ | $0.42 \pm 0.36$ | $0.22 \pm 0.08$ | $0.22 \pm 0.08$ | $0.22 \pm 0.08$ | $0.22 \pm 0.08$ | $0.00 \pm 0.00$ |
| $\rho = 0.49$ | $0.46 \pm 0.38$ | $0.39 \pm 0.23$ | $0.39 \pm 0.23$ | $0.39 \pm 0.23$ | $0.39 \pm 0.23$ | $0.34 \pm 0.48$ |

Table 4: Results on `LS10` dataset. Reported is the mean and standard deviation of the 0-1 error over 125 trials. Grayed cells denote the best performer at that noise rate.

|  | **Hinge** | **Logistic** | **Square** | $t$-**logistic** | **TanBoost** | **Unhinged** |
|---|---|---|---|---|---|---|
| $\rho = 0$ | $0.02 \pm 0.00$ | $0.01 \pm 0.00$ | $0.03 \pm 0.00$ | $0.01 \pm 0.00$ | $0.02 \pm 0.00$ | $0.05 \pm 0.00$ |
| $\rho = 0.1$ | $0.13 \pm 0.01$ | $0.05 \pm 0.01$ | $0.06 \pm 0.01$ | $0.03 \pm 0.01$ | $0.05 \pm 0.01$ | $0.06 \pm 0.01$ |
| $\rho = 0.2$ | $0.14 \pm 0.01$ | $0.09 \pm 0.02$ | $0.09 \pm 0.02$ | $0.06 \pm 0.02$ | $0.08 \pm 0.02$ | $0.08 \pm 0.02$ |
| $\rho = 0.3$ | $0.15 \pm 0.01$ | $0.13 \pm 0.03$ | $0.13 \pm 0.03$ | $0.12 \pm 0.03$ | $0.12 \pm 0.03$ | $0.12 \pm 0.02$ |
| $\rho = 0.4$ | $0.17 \pm 0.05$ | $0.24 \pm 0.08$ | $0.24 \pm 0.08$ | $0.23 \pm 0.07$ | $0.23 \pm 0.08$ | $0.23 \pm 0.08$ |
| $\rho = 0.49$ | $0.47 \pm 0.24$ | $0.46 \pm 0.11$ | $0.47 \pm 0.11$ | $0.48 \pm 0.10$ | $0.47 \pm 0.12$ | $0.48 \pm 0.12$ |

(a) 0-1 Error.

|  | **Hinge** | **Logistic** | **Square** | $t$-**logistic** | **TanBoost** | **Unhinged** |
|---|---|---|---|---|---|---|
| $\rho = 0$ | $0.00 \pm 0.00$ | $0.00 \pm 0.00$ | $0.01 \pm 0.00$ | $0.00 \pm 0.00$ | $0.00 \pm 0.00$ | $0.01 \pm 0.00$ |
| $\rho = 0.1$ | $0.25 \pm 0.10$ | $0.02 \pm 0.01$ | $0.02 \pm 0.01$ | $0.00 \pm 0.00$ | $0.02 \pm 0.01$ | $0.02 \pm 0.01$ |
| $\rho = 0.2$ | $0.34 \pm 0.10$ | $0.05 \pm 0.02$ | $0.05 \pm 0.02$ | $0.02 \pm 0.01$ | $0.04 \pm 0.02$ | $0.05 \pm 0.02$ |
| $\rho = 0.3$ | $0.41 \pm 0.11$ | $0.11 \pm 0.04$ | $0.11 \pm 0.04$ | $0.09 \pm 0.04$ | $0.11 \pm 0.04$ | $0.10 \pm 0.04$ |
| $\rho = 0.4$ | $0.44 \pm 0.12$ | $0.24 \pm 0.08$ | $0.24 \pm 0.08$ | $0.24 \pm 0.08$ | $0.24 \pm 0.08$ | $0.23 \pm 0.08$ |
| $\rho = 0.49$ | $0.50 \pm 0.13$ | $0.47 \pm 0.11$ | $0.47 \pm 0.11$ | $0.47 \pm 0.11$ | $0.47 \pm 0.11$ | $0.46 \pm 0.11$ |

(b) 1 - AUC.

Table 5: Results on `mease` dataset. Reported is the mean and standard deviation of performance over 125 trials. Grayed cells denote the best performer at that noise rate.

|  | **Hinge** | **Logistic** | **Square** | *t*-**logistic** | **TanBoost** | **Unhinged** |
|---|---|---|---|---|---|---|
| $\rho = 0$ | $0.00 \pm 0.00$ | $0.00 \pm 0.00$ | $0.00 \pm 0.00$ | $0.00 \pm 0.00$ | $0.00 \pm 0.00$ | $0.00 \pm 0.00$ |
| $\rho = 0.1$ | $0.01 \pm 0.03$ | $0.01 \pm 0.01$ | $0.01 \pm 0.02$ | $0.01 \pm 0.03$ | $0.01 \pm 0.02$ | $0.00 \pm 0.00$ |
| $\rho = 0.2$ | $0.06 \pm 0.12$ | $0.02 \pm 0.05$ | $0.03 \pm 0.04$ | $0.04 \pm 0.05$ | $0.03 \pm 0.05$ | $0.00 \pm 0.01$ |
| $\rho = 0.3$ | $0.17 \pm 0.20$ | $0.09 \pm 0.10$ | $0.08 \pm 0.09$ | $0.09 \pm 0.11$ | $0.09 \pm 0.10$ | $0.02 \pm 0.07$ |
| $\rho = 0.4$ | $0.35 \pm 0.24$ | $0.24 \pm 0.17$ | $0.24 \pm 0.17$ | $0.24 \pm 0.16$ | $0.24 \pm 0.17$ | $0.13 \pm 0.22$ |
| $\rho = 0.49$ | $0.60 \pm 0.20$ | $0.49 \pm 0.20$ | $0.49 \pm 0.19$ | $0.49 \pm 0.20$ | $0.49 \pm 0.19$ | $0.45 \pm 0.33$ |

(a) 0-1 Error.

|  | **Hinge** | **Logistic** | **Square** | *t*-**logistic** | **TanBoost** | **Unhinged** |
|---|---|---|---|---|---|---|
| $\rho = 0$ | $0.00 \pm 0.00$ | $0.00 \pm 0.00$ | $0.00 \pm 0.00$ | $0.00 \pm 0.00$ | $0.00 \pm 0.00$ | $0.00 \pm 0.00$ |
| $\rho = 0.1$ | $0.00 \pm 0.00$ | $0.00 \pm 0.00$ | $0.00 \pm 0.00$ | $0.00 \pm 0.00$ | $0.00 \pm 0.00$ | $0.00 \pm 0.00$ |
| $\rho = 0.2$ | $0.03 \pm 0.11$ | $0.00 \pm 0.01$ | $0.00 \pm 0.00$ | $0.00 \pm 0.01$ | $0.00 \pm 0.01$ | $0.00 \pm 0.00$ |
| $\rho = 0.3$ | $0.14 \pm 0.26$ | $0.02 \pm 0.06$ | $0.02 \pm 0.05$ | $0.02 \pm 0.06$ | $0.02 \pm 0.05$ | $0.01 \pm 0.06$ |
| $\rho = 0.4$ | $0.36 \pm 0.38$ | $0.13 \pm 0.18$ | $0.13 \pm 0.18$ | $0.14 \pm 0.18$ | $0.13 \pm 0.18$ | $0.09 \pm 0.27$ |
| $\rho = 0.49$ | $0.72 \pm 0.34$ | $0.47 \pm 0.31$ | $0.48 \pm 0.30$ | $0.48 \pm 0.30$ | $0.48 \pm 0.30$ | $0.45 \pm 0.48$ |

(b) 1 - AUC.

Table 6: Results on `iris` dataset. Reported is the mean and standard deviation of performance over 125 trials. Grayed cells denote the best performer at that noise rate.

|  | **Hinge** | **Logistic** | **Square** | *t*-**logistic** | **TanBoost** | **Unhinged** |
|---|---|---|---|---|---|---|
| $\rho = 0$ | $0.11 \pm 0.00$ | $0.13 \pm 0.00$ | $0.17 \pm 0.00$ | $0.24 \pm 0.00$ | $0.17 \pm 0.00$ | $0.20 \pm 0.00$ |
| $\rho = 0.1$ | $0.17 \pm 0.04$ | $0.18 \pm 0.04$ | $0.16 \pm 0.03$ | $0.19 \pm 0.05$ | $0.17 \pm 0.04$ | $0.19 \pm 0.02$ |
| $\rho = 0.2$ | $0.20 \pm 0.05$ | $0.19 \pm 0.05$ | $0.18 \pm 0.04$ | $0.21 \pm 0.06$ | $0.18 \pm 0.04$ | $0.19 \pm 0.02$ |
| $\rho = 0.3$ | $0.23 \pm 0.06$ | $0.22 \pm 0.05$ | $0.22 \pm 0.05$ | $0.24 \pm 0.06$ | $0.22 \pm 0.05$ | $0.21 \pm 0.03$ |
| $\rho = 0.4$ | $0.31 \pm 0.11$ | $0.31 \pm 0.10$ | $0.29 \pm 0.09$ | $0.32 \pm 0.09$ | $0.30 \pm 0.10$ | $0.27 \pm 0.12$ |
| $\rho = 0.49$ | $0.48 \pm 0.16$ | $0.47 \pm 0.16$ | $0.47 \pm 0.16$ | $0.47 \pm 0.14$ | $0.45 \pm 0.15$ | $0.46 \pm 0.22$ |

(a) 0-1 Error.

|  | **Hinge** | **Logistic** | **Square** | *t*-**logistic** | **TanBoost** | **Unhinged** |
|---|---|---|---|---|---|---|
| $\rho = 0$ | $0.12 \pm 0.00$ | $0.13 \pm 0.00$ | $0.07 \pm 0.00$ | $0.20 \pm 0.00$ | $0.07 \pm 0.00$ | $0.21 \pm 0.00$ |
| $\rho = 0.1$ | $0.18 \pm 0.07$ | $0.18 \pm 0.07$ | $0.12 \pm 0.04$ | $0.22 \pm 0.07$ | $0.13 \pm 0.05$ | $0.21 \pm 0.00$ |
| $\rho = 0.2$ | $0.23 \pm 0.09$ | $0.22 \pm 0.09$ | $0.18 \pm 0.07$ | $0.25 \pm 0.08$ | $0.19 \pm 0.08$ | $0.21 \pm 0.01$ |
| $\rho = 0.3$ | $0.31 \pm 0.11$ | $0.29 \pm 0.09$ | $0.26 \pm 0.09$ | $0.30 \pm 0.09$ | $0.27 \pm 0.09$ | $0.21 \pm 0.01$ |
| $\rho = 0.4$ | $0.40 \pm 0.11$ | $0.40 \pm 0.10$ | $0.38 \pm 0.10$ | $0.40 \pm 0.10$ | $0.38 \pm 0.10$ | $0.25 \pm 0.12$ |
| $\rho = 0.49$ | $0.49 \pm 0.12$ | $0.50 \pm 0.10$ | $0.50 \pm 0.10$ | $0.50 \pm 0.10$ | $0.50 \pm 0.10$ | $0.46 \pm 0.25$ |

(b) 1 - AUC.

Table 7: Results on `ionosphere` dataset. Reported is the mean and standard deviation of performance over 125 trials. Grayed cells denote the best performer at that noise rate.

|  | **Hinge** | **Logistic** | **Square** | *t*-**logistic** | **TanBoost** | **Unhinged** |
|---|---|---|---|---|---|---|
| $\rho = 0$ | $0.05 \pm 0.00$ | $0.05 \pm 0.00$ | $0.07 \pm 0.00$ | $0.05 \pm 0.00$ | $0.07 \pm 0.00$ | $0.05 \pm 0.00$ |
| $\rho = 0.1$ | $0.06 \pm 0.01$ | $0.06 \pm 0.02$ | $0.07 \pm 0.02$ | $0.07 \pm 0.02$ | $0.07 \pm 0.02$ | $0.05 \pm 0.00$ |
| $\rho = 0.2$ | $0.06 \pm 0.01$ | $0.07 \pm 0.03$ | $0.07 \pm 0.02$ | $0.08 \pm 0.03$ | $0.07 \pm 0.02$ | $0.05 \pm 0.00$ |
| $\rho = 0.3$ | $0.08 \pm 0.04$ | $0.10 \pm 0.06$ | $0.11 \pm 0.06$ | $0.11 \pm 0.05$ | $0.11 \pm 0.06$ | $0.05 \pm 0.01$ |
| $\rho = 0.4$ | $0.14 \pm 0.10$ | $0.21 \pm 0.12$ | $0.22 \pm 0.12$ | $0.24 \pm 0.13$ | $0.22 \pm 0.13$ | $0.09 \pm 0.10$ |
| $\rho = 0.49$ | $0.45 \pm 0.26$ | $0.49 \pm 0.16$ | $0.50 \pm 0.16$ | $0.49 \pm 0.16$ | $0.51 \pm 0.17$ | $0.46 \pm 0.30$ |

(a) 0-1 Error.

|  | **Hinge** | **Logistic** | **Square** | *t*-**logistic** | **TanBoost** | **Unhinged** |
|---|---|---|---|---|---|---|
| $\rho = 0$ | $0.25 \pm 0.00$ | $0.15 \pm 0.00$ | $0.17 \pm 0.00$ | $0.25 \pm 0.00$ | $0.17 \pm 0.00$ | $0.69 \pm 0.00$ |
| $\rho = 0.1$ | $0.38 \pm 0.12$ | $0.27 \pm 0.07$ | $0.27 \pm 0.07$ | $0.30 \pm 0.09$ | $0.27 \pm 0.07$ | $0.69 \pm 0.00$ |
| $\rho = 0.2$ | $0.41 \pm 0.13$ | $0.35 \pm 0.10$ | $0.35 \pm 0.10$ | $0.35 \pm 0.10$ | $0.35 \pm 0.10$ | $0.68 \pm 0.00$ |
| $\rho = 0.3$ | $0.44 \pm 0.12$ | $0.40 \pm 0.11$ | $0.40 \pm 0.11$ | $0.40 \pm 0.11$ | $0.40 \pm 0.11$ | $0.69 \pm 0.01$ |
| $\rho = 0.4$ | $0.43 \pm 0.12$ | $0.45 \pm 0.12$ | $0.45 \pm 0.12$ | $0.45 \pm 0.12$ | $0.45 \pm 0.12$ | $0.68 \pm 0.02$ |
| $\rho = 0.49$ | $0.45 \pm 0.13$ | $0.49 \pm 0.13$ | $0.49 \pm 0.13$ | $0.49 \pm 0.13$ | $0.49 \pm 0.13$ | $0.57 \pm 0.16$ |

(b) 1 - AUC.

Table 8: Results on `housing` dataset. Reported is the mean and standard deviation of performance over 125 trials. Grayed cells denote the best performer at that noise rate.

|  | **Hinge** | **Logistic** | **Square** | *t*-**logistic** | **TanBoost** | **Unhinged** |
|---|---|---|---|---|---|---|
| $\rho = 0$ | $0.01 \pm 0.00$ | $0.02 \pm 0.00$ | $0.03 \pm 0.00$ | $0.03 \pm 0.00$ | $0.02 \pm 0.00$ | $0.03 \pm 0.00$ |
| $\rho = 0.1$ | $0.05 \pm 0.00$ | $0.04 \pm 0.01$ | $0.04 \pm 0.01$ | $0.02 \pm 0.01$ | $0.04 \pm 0.01$ | $0.04 \pm 0.01$ |
| $\rho = 0.2$ | $0.05 \pm 0.00$ | $0.05 \pm 0.01$ | $0.05 \pm 0.01$ | $0.04 \pm 0.01$ | $0.05 \pm 0.01$ | $0.05 \pm 0.01$ |
| $\rho = 0.3$ | $0.05 \pm 0.01$ | $0.06 \pm 0.01$ | $0.06 \pm 0.01$ | $0.06 \pm 0.02$ | $0.06 \pm 0.01$ | $0.06 \pm 0.01$ |
| $\rho = 0.4$ | $0.06 \pm 0.02$ | $0.11 \pm 0.06$ | $0.11 \pm 0.06$ | $0.11 \pm 0.06$ | $0.11 \pm 0.06$ | $0.10 \pm 0.05$ |
| $\rho = 0.49$ | $0.33 \pm 0.27$ | $0.46 \pm 0.16$ | $0.46 \pm 0.16$ | $0.47 \pm 0.16$ | $0.47 \pm 0.16$ | $0.46 \pm 0.16$ |

(a) 0-1 Error.

|  | **Hinge** | **Logistic** | **Square** | *t*-**logistic** | **TanBoost** | **Unhinged** |
|---|---|---|---|---|---|---|
| $\rho = 0$ | $0.00 \pm 0.00$ | $0.00 \pm 0.00$ | $0.01 \pm 0.00$ | $0.00 \pm 0.00$ | $0.01 \pm 0.00$ | $0.02 \pm 0.00$ |
| $\rho = 0.1$ | $0.34 \pm 0.18$ | $0.03 \pm 0.02$ | $0.03 \pm 0.02$ | $0.00 \pm 0.00$ | $0.03 \pm 0.02$ | $0.04 \pm 0.02$ |
| $\rho = 0.2$ | $0.40 \pm 0.17$ | $0.07 \pm 0.05$ | $0.08 \pm 0.05$ | $0.04 \pm 0.04$ | $0.07 \pm 0.05$ | $0.08 \pm 0.05$ |
| $\rho = 0.3$ | $0.43 \pm 0.17$ | $0.17 \pm 0.10$ | $0.17 \pm 0.10$ | $0.14 \pm 0.10$ | $0.16 \pm 0.10$ | $0.16 \pm 0.10$ |
| $\rho = 0.4$ | $0.44 \pm 0.18$ | $0.30 \pm 0.16$ | $0.30 \pm 0.16$ | $0.30 \pm 0.16$ | $0.30 \pm 0.16$ | $0.30 \pm 0.16$ |
| $\rho = 0.49$ | $0.51 \pm 0.19$ | $0.46 \pm 0.17$ | $0.46 \pm 0.17$ | $0.46 \pm 0.17$ | $0.46 \pm 0.17$ | $0.46 \pm 0.18$ |

(b) 1 - AUC.

Table 9: Results on `car` dataset. Reported is the mean and standard deviation of performance over 125 trials. Grayed cells denote the best performer at that noise rate.

|  | **Hinge** | **Logistic** | **Square** | **$t$-logistic** | **TanBoost** | **Unhinged** |
|---|---|---|---|---|---|---|
| $\rho = 0$ | $0.00 \pm 0.00$ | $0.00 \pm 0.00$ | $0.00 \pm 0.00$ | $0.00 \pm 0.00$ | $0.00 \pm 0.00$ | $0.00 \pm 0.00$ |
| $\rho = 0.1$ | $0.10 \pm 0.08$ | $0.05 \pm 0.01$ | $0.01 \pm 0.01$ | $0.11 \pm 0.02$ | $0.02 \pm 0.01$ | $0.00 \pm 0.00$ |
| $\rho = 0.2$ | $0.19 \pm 0.11$ | $0.09 \pm 0.02$ | $0.05 \pm 0.02$ | $0.15 \pm 0.02$ | $0.06 \pm 0.02$ | $0.00 \pm 0.00$ |
| $\rho = 0.3$ | $0.31 \pm 0.13$ | $0.17 \pm 0.03$ | $0.14 \pm 0.02$ | $0.22 \pm 0.03$ | $0.16 \pm 0.03$ | $0.01 \pm 0.00$ |
| $\rho = 0.4$ | $0.39 \pm 0.13$ | $0.31 \pm 0.04$ | $0.30 \pm 0.04$ | $0.33 \pm 0.04$ | $0.31 \pm 0.04$ | $0.02 \pm 0.02$ |
| $\rho = 0.49$ | $0.50 \pm 0.16$ | $0.48 \pm 0.04$ | $0.47 \pm 0.04$ | $0.48 \pm 0.04$ | $0.48 \pm 0.04$ | $0.34 \pm 0.21$ |

(a) 0-1 Error.

|  | **Hinge** | **Logistic** | **Square** | **$t$-logistic** | **TanBoost** | **Unhinged** |
|---|---|---|---|---|---|---|
| $\rho = 0$ | $0.00 \pm 0.00$ | $0.00 \pm 0.00$ | $0.00 \pm 0.00$ | $0.00 \pm 0.00$ | $0.00 \pm 0.00$ | $0.00 \pm 0.00$ |
| $\rho = 0.1$ | $0.05 \pm 0.06$ | $0.01 \pm 0.00$ | $0.00 \pm 0.00$ | $0.05 \pm 0.01$ | $0.00 \pm 0.00$ | $0.00 \pm 0.00$ |
| $\rho = 0.2$ | $0.12 \pm 0.11$ | $0.03 \pm 0.01$ | $0.01 \pm 0.00$ | $0.07 \pm 0.01$ | $0.02 \pm 0.01$ | $0.00 \pm 0.00$ |
| $\rho = 0.3$ | $0.26 \pm 0.18$ | $0.10 \pm 0.02$ | $0.07 \pm 0.02$ | $0.14 \pm 0.03$ | $0.08 \pm 0.02$ | $0.00 \pm 0.00$ |
| $\rho = 0.4$ | $0.37 \pm 0.19$ | $0.25 \pm 0.04$ | $0.24 \pm 0.04$ | $0.27 \pm 0.04$ | $0.24 \pm 0.04$ | $0.00 \pm 0.00$ |
| $\rho = 0.49$ | $0.51 \pm 0.23$ | $0.47 \pm 0.05$ | $0.46 \pm 0.05$ | $0.47 \pm 0.05$ | $0.47 \pm 0.05$ | $0.25 \pm 0.29$ |

(b) 1 - AUC.

Table 10: Results on `usps_0_vs_7` dataset. Reported is the mean and standard deviation of performance over 125 trials. Grayed cells denote the best performer at that noise rate.

|  | **Hinge** | **Logistic** | **Square** | **$t$-logistic** | **TanBoost** | **Unhinged** |
|---|---|---|---|---|---|---|
| $\rho = 0$ | $0.05 \pm 0.00$ | $0.04 \pm 0.00$ | $0.02 \pm 0.00$ | $0.04 \pm 0.00$ | $0.02 \pm 0.00$ | $0.19 \pm 0.00$ |
| $\rho = 0.1$ | $0.15 \pm 0.03$ | $0.05 \pm 0.01$ | $0.04 \pm 0.01$ | $0.24 \pm 0.00$ | $0.04 \pm 0.01$ | $0.19 \pm 0.01$ |
| $\rho = 0.2$ | $0.21 \pm 0.03$ | $0.08 \pm 0.01$ | $0.07 \pm 0.01$ | $0.24 \pm 0.00$ | $0.07 \pm 0.01$ | $0.19 \pm 0.01$ |
| $\rho = 0.3$ | $0.25 \pm 0.03$ | $0.14 \pm 0.02$ | $0.14 \pm 0.02$ | $0.24 \pm 0.00$ | $0.14 \pm 0.02$ | $0.19 \pm 0.03$ |
| $\rho = 0.4$ | $0.31 \pm 0.05$ | $0.28 \pm 0.05$ | $0.28 \pm 0.04$ | $0.24 \pm 0.00$ | $0.28 \pm 0.04$ | $0.22 \pm 0.05$ |
| $\rho = 0.49$ | $0.48 \pm 0.09$ | $0.47 \pm 0.06$ | $0.48 \pm 0.05$ | $0.40 \pm 0.24$ | $0.48 \pm 0.05$ | $0.45 \pm 0.08$ |

(a) 0-1 Error.

|  | **Hinge** | **Logistic** | **Square** | **$t$-logistic** | **TanBoost** | **Unhinged** |
|---|---|---|---|---|---|---|
| $\rho = 0$ | $0.01 \pm 0.00$ | $0.01 \pm 0.00$ | $0.00 \pm 0.00$ | $0.01 \pm 0.00$ | $0.00 \pm 0.00$ | $0.09 \pm 0.00$ |
| $\rho = 0.1$ | $0.10 \pm 0.03$ | $0.01 \pm 0.00$ | $0.01 \pm 0.00$ | $0.03 \pm 0.01$ | $0.01 \pm 0.00$ | $0.09 \pm 0.01$ |
| $\rho = 0.2$ | $0.20 \pm 0.05$ | $0.03 \pm 0.01$ | $0.02 \pm 0.01$ | $0.04 \pm 0.01$ | $0.02 \pm 0.01$ | $0.10 \pm 0.02$ |
| $\rho = 0.3$ | $0.30 \pm 0.06$ | $0.08 \pm 0.02$ | $0.08 \pm 0.02$ | $0.09 \pm 0.02$ | $0.07 \pm 0.02$ | $0.11 \pm 0.03$ |
| $\rho = 0.4$ | $0.40 \pm 0.07$ | $0.22 \pm 0.04$ | $0.22 \pm 0.04$ | $0.23 \pm 0.04$ | $0.22 \pm 0.04$ | $0.16 \pm 0.07$ |
| $\rho = 0.49$ | $0.49 \pm 0.08$ | $0.46 \pm 0.05$ | $0.46 \pm 0.05$ | $0.46 \pm 0.05$ | $0.45 \pm 0.05$ | $0.42 \pm 0.15$ |

(b) 1 - AUC.

Table 11: Results on `splice` dataset. Reported is the mean and standard deviation of performance over 125 trials. Grayed cells denote the best performer at that noise rate.

|  | **Hinge** | **Logistic** | **Square** | *t*-**logistic** | **TanBoost** | **Unhinged** |
|---|---|---|---|---|---|---|
| $\rho = 0$ | $0.16 \pm 0.01$ | $0.08 \pm 0.00$ | $0.10 \pm 0.00$ | $0.24 \pm 0.00$ | $0.09 \pm 0.00$ | $0.15 \pm 0.00$ |
| $\rho = 0.1$ | $0.14 \pm 0.03$ | $0.10 \pm 0.02$ | $0.10 \pm 0.01$ | $0.13 \pm 0.06$ | $0.10 \pm 0.01$ | $0.14 \pm 0.01$ |
| $\rho = 0.2$ | $0.17 \pm 0.03$ | $0.11 \pm 0.02$ | $0.11 \pm 0.01$ | $0.13 \pm 0.05$ | $0.11 \pm 0.01$ | $0.14 \pm 0.01$ |
| $\rho = 0.3$ | $0.23 \pm 0.05$ | $0.13 \pm 0.02$ | $0.12 \pm 0.01$ | $0.14 \pm 0.04$ | $0.13 \pm 0.02$ | $0.15 \pm 0.01$ |
| $\rho = 0.4$ | $0.33 \pm 0.07$ | $0.20 \pm 0.04$ | $0.19 \pm 0.03$ | $0.21 \pm 0.04$ | $0.19 \pm 0.03$ | $0.17 \pm 0.03$ |
| $\rho = 0.49$ | $0.49 \pm 0.10$ | $0.45 \pm 0.07$ | $0.44 \pm 0.07$ | $0.45 \pm 0.07$ | $0.45 \pm 0.07$ | $0.43 \pm 0.12$ |

(a) 0-1 Error.

|  | **Hinge** | **Logistic** | **Square** | *t*-**logistic** | **TanBoost** | **Unhinged** |
|---|---|---|---|---|---|---|
| $\rho = 0$ | $0.03 \pm 0.00$ | $0.02 \pm 0.00$ | $0.05 \pm 0.00$ | $0.02 \pm 0.00$ | $0.04 \pm 0.00$ | $0.07 \pm 0.00$ |
| $\rho = 0.1$ | $0.06 \pm 0.01$ | $0.04 \pm 0.00$ | $0.05 \pm 0.00$ | $0.03 \pm 0.00$ | $0.04 \pm 0.00$ | $0.07 \pm 0.00$ |
| $\rho = 0.2$ | $0.10 \pm 0.03$ | $0.05 \pm 0.00$ | $0.05 \pm 0.00$ | $0.04 \pm 0.00$ | $0.05 \pm 0.00$ | $0.07 \pm 0.00$ |
| $\rho = 0.3$ | $0.17 \pm 0.06$ | $0.06 \pm 0.01$ | $0.06 \pm 0.01$ | $0.06 \pm 0.01$ | $0.06 \pm 0.01$ | $0.07 \pm 0.01$ |
| $\rho = 0.4$ | $0.32 \pm 0.12$ | $0.12 \pm 0.02$ | $0.12 \pm 0.02$ | $0.12 \pm 0.02$ | $0.12 \pm 0.02$ | $0.09 \pm 0.02$ |
| $\rho = 0.49$ | $0.49 \pm 0.14$ | $0.43 \pm 0.08$ | $0.43 \pm 0.08$ | $0.43 \pm 0.07$ | $0.43 \pm 0.08$ | $0.39 \pm 0.19$ |

(b) 1 - AUC.

Table 12: Results on spambase dataset. Reported is the mean and standard deviation of performance over 125 trials. Grayed cells denote the best performer at that noise rate.

## Footnotes

[6]The specific choice of $D$ *requires* that one not include a bias term; with a bias term, it can be checked that the example as-stated has a trivial solution.

[7]The result also holds if we add a regularised bias term. With an unregularised bias term, Bedo et al. [2006] showed that the limiting solution of a soft-margin SVM is distribution dependent.

[8]Another interesting observation is that these noise-corrected losses are negatively unbounded – that is, minimising hinge loss on $D$ is equivalent to minimising a negatively unbounded loss on $\bar{D}$. This is another justification for studying negatively unbounded losses.

[9]For a general (not necessarily translation invariant) kernel, this is known as a potential function rule [Devroye et al., 1996, §10.3]. The use of "potential" here is distinct from that of a "convex potential".

[10]This refers to the rate of convergence of the estimate of $\eta$ to the true $\eta$. By contrast, generalisation bounds establish that the rate of convergence of the estimate of the corresponding *classifier* to the Bayes-optimal classifier $\mathrm{sign}(2\eta(x) - 1)$ is independent of the dimension of the feature space.