[Reviews · NeurIPS 2015]

Submitted by Assigned_Reviewer_1

SUMMARY

The paper presents a solution to binary classification with symmetric label noise (SLN). They show that, in order to obtain consistency (w.r.t. to the 0-1 loss in the "noiseless" case) while using a convex surrogate, one must use the loss $\ell(v,y) = 1 - vy$ -- the "unhinged loss" -- , which is shown to enjoy some useful properties, including robustness to SLN. In a more restricted sense of robustness, it is the only such loss, but in any case it overcomes the limitations of other convex losses for the same problem.

Different implications of using the unhinged loss are discussed; the problem of classification with SLN with the unhinged loss and "linear" classifiers is investigated and solved analytically. The authors also present an empirical evaluation to motivate that their theoretical considerations have practical impact.

CONTRIBUTIONS

The main contribution is an SLN-robust loss (consistent in spite of label noise), and a demonstration that this function enjoys certain properties that make it suitable for classification with SLN, including with linear classifiers.

The contributions are relevant for the problem of classification with SLN; they have limited scope outside the problem, which in this reviewer's opinion somewhat specific, but at the same time they address the it effectively and describe a clear-cut solution for the problem.

SUPPORT

The support for the contributions is adequate. The results are clearly stated and the proofs are correct easy to follow. The illustrative empirical support of the theoretical results is also suitable, and may shed light on the finite-sample behavior of minimizers of SLN robust losses.

TECHNICAL QUALITY

The paper is technically sound and well-executed.

A discussion of the limitations of strong SLN robustness (as opposed to SLN robustness) would be informative, as would a discussion about the limitations of the SLN problem as a model for other problems would (though somewhat non-central).

More considerations about the finite-sample behavior of minimizers of the unhinged loss would strengthen the discussion of the empirical results.

ORIGINALITY

The work is well-placed in the literature, the results well-connected to existing ones, and contributions, novel.

CLARITY

The paper is well written and easy to read. The different facts about the unhinged loss are brought together cohesively.

Considering binary classification problems in general, the problem of classification with SLN is presented as more realistic than the general case, but this reviewer would appreciate some more motivation as to why one believes so (see detailed comments). (Appendix B.1 seems to briefly touch on this matter.) Perhaps SLN robustness is a step towards robustness to some types of outliers?

FOR THE REBUTTAL

I would appreciate if the authors could give a brief motivation on SLN problems, and their importance in a more general binary classification context.

DETAILED COMMENTS

I am a bit confused as to the motivation behind SLN. On the one hand, in two dimensions it would be odd to just flip classes uniformly over the space. On the other hand, maybe this behavior in higher dimensions is not too strange and makes sense to be considered. Are there any practical issues motivating the interest in SLN (applications, robustness, etc.), or if the interest is of theoretical nature?

Although the loss is unhinged, unboundedness is dealt with by bounding the scores. This essentially turns the unhinged loss into a hinge loss again. I would suggest looking into [1,2], from which we can infer that with $\mathcal{F} = \mathbb{R}^{[-1,1]}$ the hinge and unhinged losses are equivalent and, by a scaling argument, that so are the unhinged and the "scaled" hinge $f(v) = (1 + v/B)_+$ with $\mathcal{F} = \mathbb{R}^{[-B,B]}$. In terms of regret/excess risk (i.e., ignoring constants added to the losses) one can also obtain equivalence with $f(v) = \frac{1}{B}(1 + v)_+$. These results can be used to even obtain tight risk bounds with calibration functions [3].

Regarding Conjecture 1 in Appendix B.2, I would point out that linear scorers and (some) proper losses do not go very well together: even in the absence of noise, minimizers of these losses fail to be consistent w.r.t. the best linear classifier in the class [4]. Perhaps the proof techniques used in [4] can provide some directions to develop proofs for Conjecture 1. I would also imagine that Conjecture 1 with linear scorers with $\|w\| \leq C$ for constants $C$ would be quite interesting to investigate.

Based on your experimental results, and considering that the theoretical results are asymptotic in nature, can you make any statements about SLN robustness in the finite-sample setting based on your data?

REFERENCES

[1] Zou, Zhu & Hastie, 2006: "The margin vector, admissible loss and multi-class margin based classifiers" [2] Avila Pires, Szepesvari & Ghavamzadeh, 2013: "Cost-sensitive multiclass classification risk bounds" [3] Steinwart, 2007: "How to compare different loss functions and their risks" [4] Long & Servedio, 2013: "Consistency versus Realizable $H$-Consistency for Multiclass Classification"

SUGGESTED FIXES

[l:75] "argmin" [l:120-121] "w.r.t." [l:144-145] Your appendix references seem to be broken. [l:234-237]

This intuition is compatible with a "graphical" view of the SLN setup, where we would lay out points in space and uniformly flip their labels with a given probability. This also begs the question as to whether this setup is realistic at all, of course, but it makes sense that a classifier should be able to "ignore" large errors introduced by the noise in order to succeed. [l:245] Interestingly, Proposition 5 is also the case for the hinge loss with scaled scores, which is equivalent to the unhinged loss with bounded scores. Note that you instantly get even risk bounds in that case. [l:270] In Proposition 6, you can have R_{01}(s) = B \cdot R_{\ell}(s) (by looking at the minimizers directly). [l:308-311] Because you already have the expectations, you do not need to regularize in the Ridge sense -- I was going to suggest that you can regularize by biasing the centroid estimates, but you already allude to that in the next paragraph. [l:341-347] The statement here seems like a natural consequence of over-regularization. The conclusions you use for SLN robustness are interesting, nonetheless. [l:344-345] $\limsup_{\lambda \rightarrow \infty}$? [l:362] The appendix references are broken here too. [l:445] "$\ell$-risk w.r.t. some convex" [l:480] "any $\rho < \frac{1}{2}$" [l:494] This notation where one integrates out the subscript can be confusing... I would suggest using conditional expectation. [l:514] I believe you need $\alpha \geq 0$. [l:535] $(1 - 2\eta)v + 1$

POST-REBUTTAL REMARKS

This paper has the potential to reach a larger audience on account of its proposed algorithm, which the authors should tap. Although SLN has attracted interest from the theoretical and practical points of view, I have the impression that it is too harsh a noise model. Nevertheless, this work is a step in the direction of understanding how to have robustness to noise in classification.

The lack of comparison between SLN robustness and its strong counterpart seems to have been a concern in other reviews. While strong SLN robustness ensures that a loss is suitable for classification with SLN, it also restricts the number of such suitable losses. It would be interesting to understand whether how limiting this restriction is.

The Appendix carries a relatively large portion of work that may not have been peer-reviwed, but this reviewer verified the proofs (Appendix A).
Summary: The paper is well-executed and well-structured; it manages to cover different aspects of the loss without losing focus of the task at hand, and it proposes a well-supported solution for the problem studied.

Submitted by Assigned_Reviewer_2

The paper considers the problem of learning with (symmetric) random label noise and proposes a simple convex loss that is provably robust and classification-calibrated. While there have been many recent results on this problem (which the authors adequately cite and make connections to), the paper makes several independent contributions; the key contributions are (almost) closing the gap in the analysis of a certain sufficiency condition on the loss function for noise tolerance introduced by Manwani & Sastry [2013], and complementing the results of Long & Servidio [2010] for learning with symmetric label noise.

The theory is very neatly developed in the paper. While the unhinged loss itself is rather uninteresting, the authors establish many interesting theoretical properties. In particular, the loss is shown to be calibrated with respect to the 0-1 loss. On the other hand, it is interesting to note that unhinging the standard hinge loss is not a coincidence --- the paper shows that any loss that is (strongly) SLN-robust has to necessarily take the form (up to inconsequential changes) of the proposed loss. Experimental results are sufficient (appreciate the authors for presenting elaborate results in Appendix; the results suggest that unhinging indeed helps at high noise levels). One thing that is not clear from the way the authors have presented in the main sections is if & how exactly the unhinged loss "overcomes" the statistical-computational tradeoff of using the standard margin-based losses that are SLN-robust (the authors do raise this point in the beginning, but I'm not sure if they revisited this aspect towards the end).

Overall the paper is strong and I tend to accept. It would be interesting to see how the analysis extends to the case of asymmetric noise.

Summary: The authors propose a simple unhinged loss (as against the familiar hinge loss) that is provably robust to symmetric label noise in binary classification. Several theoretical results are established regarding the unhinged loss (esp. classification-calibration). Experimental results are sufficient. In all, the paper is strong.

Submitted by Assigned_Reviewer_3

Learning with symmetric label noise is an important problem of practical interest. This work presents an analysis of this problem from the loss function perspective that suggests the use of the unhinged loss by virtue of its negative unboundedness. The authors

show that it is the unique strongly SLN robust convex loss. The unhinged loss is convex though not a convex potential. Therefore, compared to alternative approaches for solving this problem, it enjoys the ease of optimization of convex losses while not suffering from high sample complexity as is the case with universal kernels for example. Authors also show that it is classification calibrated and consistent when minimizing on the corrupted distribution if restricted to regularized scorers. The paper is very well written and several connections are established with other existing approaches.

The main novelty in the paper is the SLN robustness of unhinged loss and its uniqueness.

The uniqueness is not surprising though. the connections to highly regularized SVMs and centroid classifiers are interesting too. For the former, this was known in practice even though there was no theory backing that intuition. Hence, the presented result bridges the gap. It is also interesting (though not surprising) to see that the estimators on the clean and corrupted distributions only differ by a scaling. The first four pages of the paper are devoted to present previous material that lay foundations for the subsequent analysis. this is probably too much. As far as practical considerations are concerned, I do not see the interest of using a kernelized scorer rather than directly using kernel-based alternatives such as Random Fourrier Features or the Nystrom method that are scalable and in the limit, are also SLN robust. As announced in the abstract, I was expecting a more general result on consistency than the one based on enforcing boundedness of the unhinged loss via a restriction to bounded scorers. It is still an interesting result however.

Overall this paper presents a well-presented work on an interesting topic. The first three sections could be shortened to include some of the important proofs in the manuscript.

I recommend acceptance.
Summary: Overall this is an interesting and well-presented work. There is not as much novelty as claimed in the abstract. The first three sections could be shortened to include some of the important proofs in the manuscript.

Author Feedback
Author rebuttal: Thanks to the reviewers for several valuable comments.

R1:

> motivation on SLN

We believe SLN learning is of theoretical and practical interest. On the theoretical side, SLN learning is the arguably the simplest possible noise model, and it is known to break popular learners (the Long & Servedio result). The large literature on this problem suggests that a thorough understanding of what is possible is of interest to theoreticians. On the practical side, SLN learning seems a reasonable first-order approximation to practical instantiations of binary classification with unreliable ground truth. A canonical motivating example is where labels are provided by error-prone human annotators. More complex noise models (instance outliers, errors proportional to "hardness", etc.) may of course be a better representation, but understanding what is possible in the SLN case seems a necessary first step (as the reviewer suggests).

We can add some more discussion on this, as well on the distinction between strong and plain SLN-robustness.

> finite-sample behavior

We presented a Rademacher bound in Appendix D, noted briefly after Prop 6. Unsurprisingly, given the connection to hinge loss, we inherit identical estimation rates to the SVM. This also suggests that when tighter bounds are possible for the SVM, similar results will hold for the unhinged loss (as per reviewers' note on calibration functions).

We also note that the 1/(1 - 2p) term in the regret bounds arises in SLN lower bounds, e.g.

Aslam and Decatur. On the sample complexity of noise-tolerant learning. Inf. Proc. Letters, 1996.

We can add a few lines' discussion on this in the body.

> unhinged with boundedness = hinge

As per Prop 7, the unhinged loss may be viewed as the hinge loss with a bound on scores. We can foreshadow this earlier, perhaps after Prop 5. The reviewer is correct that this equivalence simplifies the proofs of Prop 5 and 6, and demystifies the noted similarity with bounds for the hinge loss. Our original intent was that a "first principles" approach might allay potential concerns about the boundedness of the loss, but on consideration, exploiting the equivalence wherever possible seems prudent. We will look to do this.

> conjecture 1

The reference [4] is appreciated, and will be added in discussing the conjecture. Investigating the conjecture with a norm constraint is interesting, and we can add a note on this.

Typographic fixes will be incorporated, and are appreciated.

R2:

> stat-comp tradeoff

Our reasoning is that the unhinged loss (a) results in a tractable convex problem (compared to a non-convex loss), and (b) does not require a rich function class (compared to a universal kernel approach, as per Prop 2). We can make this more explicit after introducing the loss.

R3:

> trimming Sec 1-3

We have no issues compressing material in the first four pages; any specific suggestions of content that can be trimmed would be welcome.

> why not RFFs?

We agree that RFF/Nystrom approaches are of practical interest, as noted in Sec 6.2. We prefer operating with kernelised scorers in the development of the theory to separate statistical and computational concerns. (With RFFs for example, one would have to commit to a translation invariant kernel, which is not necessary for the robustness theory to hold.) But we can certainly stress that using RFF/Nystrom approaches do not break any of the results.

> consistency via bounding

Our consistency results indeed rely on bounding the loss, rather than some more exotic alternative. We can certainly make this more explicit in the abstract, perhaps by stressing the regularised SVM perspective.

R4:

> "incompleteness" of strong SLN-robustness

It is correct that strong SLN-robustness doesn't capture square loss. Thus, as noted in Sec 4.2, this is a stronger notion, and a characterisation of SLN-robustness remains open. Nonetheless, we believe strong SLN-robustness is a natural notion to study, as both stochastic classifiers and order-equivalence are central concepts in orthodox decision theory. We believe Prop 3 is non obvious, and interesting given the sufficiency result of (Manwani & Sastry, 2013).

R5:

> "Average" algorithm of Servedio

The Servedio reference is appreciated, and will be added (along with some citing papers, such as Kalai et al. FOCS 2005). The algorithm is indeed essentially the same as the unhinged classifier of Eqn 9. Interestingly, while that paper operates in a more general noise model, its results require a strong assumption on the marginal distribution over instances, unlike ours. Also it is unclear whether the results extend beyond the linear kernel case. But this does raise interesting questions as to what more can be said about the unhinged loss' performance in other noise models, which seem worth exploring in future work.

R6:

We are glad we conveyed our intended message clearly!